# Key Factors Controlling Primary Production and Cyanobacterial Harmful Algal Blooms (cHABs) in a Continuous Weir System in the Nakdong River, Korea

**Jisoo Choi [1], Jun Oh Min [1], Bohyung Choi [1,2], Dokyun Kim [1], Jae Joong Kang [3], Sang Heon Lee [3], Kwangsoon Choi [4], Heesuk Lee [4], Jinyoung Jung [5] and Kyung-Hoon Shin [1,*]**

[1] Department of Marine Sciences and Convergent Technology, Hanyang University, Ansan 15588, Korea; hd48330@hanyang.ac.kr (J.C.); jomin8310@gmail.com (J.O.M.); chboh@hanyang.ac.kr (B.C.); dokyunkim@hanyang.ac.kr (D.K.)
[2] Fisheries Science Institute, Chonnam National University, Yeosu 59626, Korea
[3] Department of Oceanography, Pusan National University, Pusan 46241, Korea; jaejung@pusan.ac.kr (J.J.K.); sanglee@pusan.ac.kr (S.H.L.)
[4] K-water Institute, Daejeon 34045, Korea; kchoi@kwater.or.kr (K.C.); heesuklee@gmail.com (H.L.)
[5] Division of Polar Ocean Sciences, Korea Polar Research Institute (KOPRI), Incheon 211990, Korea; jinyoungjung@kopri.re.kr
\* Correspondence: shinkh@hanyang.ac.kr; Tel.: +82-31-400-5536; Fax: +82-31-416-6173

**Abstract:** To identify key factors that control primary production (P.P.) and trigger cyanobacterial harmful algal blooms (cHABs), we investigated spatio-temporal variations in P.P. in a continuous weir system in the Nakdong River once or twice a month from April to October 2018. P.P. was measured through an in-situ incubation experiment using a $^{13}$C tracer. Relative proportion of pigment-based phytoplankton composition was calculated by the CHEMTAX program based on pigment analysis using a high-performance liquid chromatography (HPLC). P.P. was higher in spring ($1130 \pm 1140$ mg C m$^{-2}$ d$^{-1}$) and summer ($1060 \pm 814$ mg C m$^{-2}$ d$^{-1}$) than autumn ($180 \pm 220$ mg C m$^{-2}$ d$^{-1}$), and tended to increase downstream. P.P. was negatively related to PO$_4$$^{3-}$ ($r = -0.41$, $p < 0.01$) due to utilization by phytoplankton during the spring and summer when it was high. The relative proportion of pigment-based cyanobacteria (mainly *Microcystis* sp.) was positively correlated with water temperature ($r = 0.79$, $p < 0.01$) and hydraulic retention time (HRT, $r = 0.67$, $p < 0.01$), suggesting that these two factors should affect cHABs in summer. Therefore, to control HRT could be one of the solutions for reducing cHABs in a continuous weir system.

**Keywords:** continuous weir; cHABs; primary productivity; environmental factors; pigment; CHEMTAX

## 1. Introduction

Primary productivity is controlled by various physical, chemical, and biological factors [1]. Previous studies demonstrated that primary productivity can be affected by light intensity, water temperature, hydraulic retention time (HRT), and nutrients [2–9]. Phytoplankton community composition can be also influenced by these factors [3,10,11]. Phytoplankton could contribute extremely to primary productivity in aquatic environments where the distributions of attached algae and aquatic plants are limited due to water-level fluctuations caused by weirs or dams [1]. The significant change of the amplitude of water-level fluctuations, causing strong disturbances in aquatic environments [12,13], such as phytoplankton community composition and densities by affecting the amount of suspended sediments, light availability, and diluting biomass [6,11].

Algal blooms deteriorate water quality by increasing organic matter and microbial activity. Microbial decomposition activity causes serious problems such as increased oxygen consumption and dissolved oxygen depletion [14]. In particular, the increase in cyanobacterial harmful algal blooms (cHABs) caused by climate change and nutrient inputs in recent decades is of global concern [4]. Proliferation of cyanobacteria changes the color of the water, creates scums and odors, and produces harmful toxins (e.g., microcystins, MCs) that are hazardous to drink and can potentially affect aquatic organisms [15,16].

Anthropogenic factors such as excessive nutrients input affected the increase in primary productivity [8,9]. In addition, climate change and long HRT consistently accelerate the overgrowth of cyanobacteria [13]. Lee et al. [5] suggested that the key factors controlling primary productivity were HRT and light conditions in Lake Chungpyeong, and phytoplankton physiological activity influenced by zooplankton grazing rate in Lake Paldang, Korea. Jia et al. [9] found that primary productivity was significantly correlated with dissolved total nitrogen, silicon/phosphorus, and dissolved inorganic carbon/phosphorus in the Gan River, and ammonium in Lake Poyang, China. Although many studies have been done to investigate primary productivity in freshwater environments, few previous studies have reported spatial and temporal variations in primary production, including cHABs, in a continuous weir system. It is important to identify the effects of anthropogenic environmental changes on phytoplankton in a continuous weir system being used as water resources. Our objectives in this study are to understand the key environmental factors controlling spatio-temporal variability in primary production, and to identify environmental conditions causing cHABs in the regulated water environment.

## 2. Materials and Methods

### 2.1. Study Area

The Nakdong River is the longest river in the South Korea with a length of 525 km and watershed area of 23,384 km$^2$. It is one of the largest water resources for drinking, industry, and agriculture in South Korea [17]. The areas surrounding the middle and downstream are densely populated, and industrial complexes are concentrated near by the Nakdong River [18]. The Korean government conducted the Four Major Rivers Project from 2009 to 2012 to solve repeated flooding and droughts [19]. Eight continuous weirs were constructed over approximately 200 km on the mainstream. After construction of the weirs, there was an increase in HRT, water level, and storage volume [20]. In addition, total phosphorus (TP) concentrations decreased noticeably from 2001 to 2016, particularly after 2013 due to reinforcement of water quality regulations [21]. Since the 1990s, diatom blooms have been reported in spring and winter, and cyanobacterial blooms in summer [22]. After the weir construction, the cell density of diatoms decreased, while cyanobacteria abundance increased [17].

This study was conducted 500 m upstream from the Gangjeong-Goryeong weir (ND-1), Dalseong weir (ND-4), Hapcheon-Changnyeong weir (ND-5), and Changnyeong-Haman weir (ND-6). Water samples were also collected at Samunjin bridge (ND-2; upstream of Dalseong weir) and Goryeong bridge (ND-3; midstream of Dalseong weir) located between the Gangjeong-Goryeong weir and Dalseong weir (Figure 1). These weirs are located in the middle and downstream of the Nakdong River. Sampling was carried out total 11 times from April to October 2018 once or twice a month.

### 2.2. Sample Colllection

Light intensity was measured at 0.5 m interval water depths from the surface to the bottom with a photosynthetically active radiation (PAR) quantum sensor (LI-1500, LI-COR, Lincoln, NE, USA), and light extinction coefficients (LEC) were calculated from PAR by depth. The 1% depth of relative light intensity was determined as the euphotic depth. Water temperature (°C) and pH were measured using a Hydrolab DS5X Multiparameter sonde (OTT Hydromet, Loveland, CO, USA). The secchi disk is a circular white disk with a diameter of 30 cm. Depths of disappearance of disk were measured to

confirm water transparencies at our sampling sites. Water samples from four water depths (100%, 50%, 12%, and 1% surface irradiances) were collected using a Van Dorn sampler. To analyze dissolved inorganic nutrients, dissolved inorganic carbon (DIC), particulate organic carbon (POC), and particulate nitrogen (PN), surface water samples were filtered using pre-combusted (450 °C, 4 h) GF/F filters (Whatman, Maidstone, UK). Filtrates for analysis of ammonium ($NH_4^+$), nitrate + nitrite ($NO_3^-$ + $NO_2^-$), phosphate ($PO_4^{3-}$), and silicate ($SiO_2$) were put into 125 mL high density polyethylene (HDPE) bottles (Nalgene, Rochester, New York) cleaned with HCl and deionized water (Aquapuri 5 series, Young In Scientific, Korea). Mercury chloride ($HgCl_2$, Sigma-Aldrich, ST Louis, MO, USA) was added to suppress microbial activity [23], and then transferred to the laboratory in a frozen state. Filtrates for DIC samples were also put into 40 mL amber vial with $HgCl_2$. Filter samples for analysis of POC and PN were stored frozen at −80 °C until further analysis. Water samples (500 mL) for pigment analysis were filtered through pre-combusted GF/F filters. Filters were wrapped in aluminum foil to prevent photolysis and then transferred to the laboratory in a frozen state. They were stored at −80 °C until analysis.

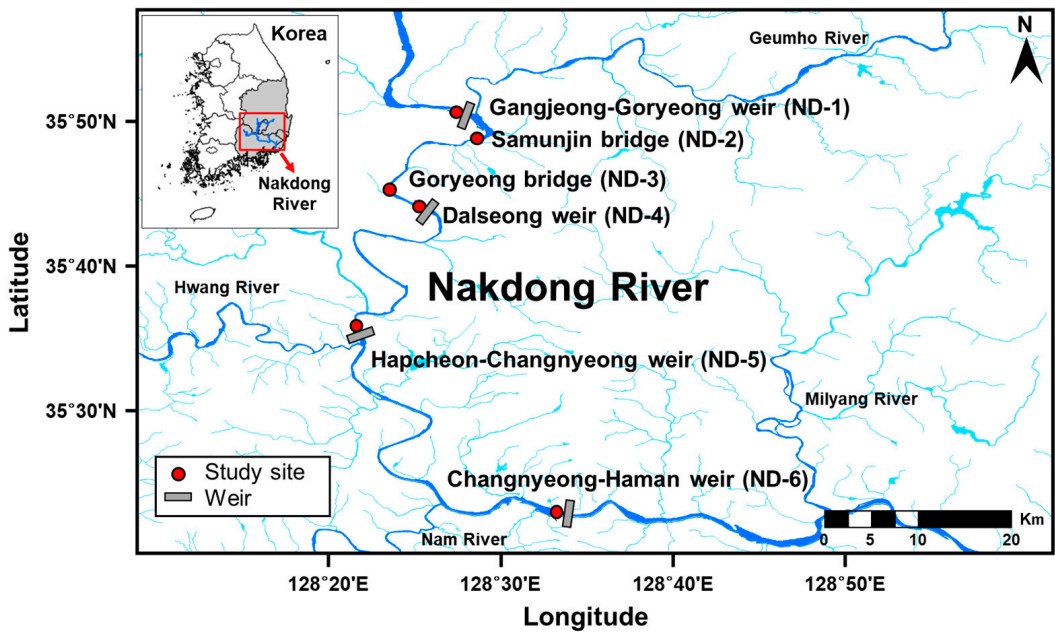

**Figure 1.** Study sites in the Nakdong River, South Korea.

### 2.3. In-situ Incubation Experiment for Primary Productivity

Primary productivity was measured thorough an in-situ incubation experiment using [13]C as a tracer. This method directly measures the amount of carbon assimilated into phytoplankton cells. In addition, in-situ incubation is allowed in the natural aquatic environments because [13]C tracer has no hazardous radioactive [24]. Water samples (500 mL) collected from each relative light depth (100%, 50%, 12%, and 1%) were filtered through a 300 μm mesh to remove mesozooplankton and transferred to polycarbonate incubation bottles (500 mL) cleaned with HCl and deionized water. The incubation bottles were covered with neutral density screens (LEE filters, Andover, UK) to match the corresponding relative light intensity (100%, 50%, 12%, and 1%) to the surface irradiances [25]. The labeled carbon stable isotope tracer (NaH[13]CO₃, 98 atom %, Sigma-Aldrich, ST Louis, MO, USA) was added to increase the [13]C atomic percent of DIC in the incubation bottles to approximately 10%, and the bottles were incubated for 4 h. After the incubation, the water samples (100 mL) were filtered through pre-combusted GF/F filters. Filters were preserved at −80 °C until analysis. Inorganic carbon was removed by 12 M HCl fuming (12 h) for freeze-dried filter samples. POC concentrations and the carbon stable isotope ratio were measured using a Finnigan Delta+XL mass spectrometer at the Alaska

Stable Isotope Facility (ASIF, University of Alaska Fairbanks, Fairbanks, USA). Carbon uptake rates of phytoplankton were calculated using the equation of Hama et al. [24].

## 2.4. Meteorological and Physicochemical Characteristics

Meteorological (precipitation) and hydrological data (inflow, discharge, and storage volume) were obtained from the Water Environment Information System (http://water.nier.go.kr). HRT was calculated at four study sites (ND-1, ND-4, ND-5, and ND-6) using the total water storage capacity and the amount of inflow water. The hydrological data from upper stream sites far from Dalseong weir (ND-2 and ND-3), which are not representative sites in front of weir, could not be shown. Nutrient concentrations were analyzed using a QuAAtro Auto-Analyzer (Seal Analytical Ltd., Southampton, UK) according to the Joint Global Ocean Flux Study (JGOFS) protocols described by Gordon et al. [26]. Concentrations of DIC were measured using a total organic carbon analyzer (TOC-L, Shimadzu, Kyoto, Japan). PN concentrations were measured with a Finnigan Delta+XL mass spectrometer at the ASIF.

## 2.5. Phytoplankton Pigment Analysis Using HPLC and Application of the CHEMTAX Program

Pigment samples lyophilized for 2 h were placed in a conical tube, and 5 mL of 100% acetone (Merck, Darmstadt, Germany) was added. Then 100 μL of canthaxanthin (Sigma-Aldrich, ST Louis, MO, USA) diluted to 2 mg L$^{-1}$ with 100% acetone was also added as the internal standard (IS). Samples were extracted in the dark at 4 °C for 24 h after ultra-sonication for 30 s. One milliliter of supernatant was filtered using a syringe filter (Polytetrafluoroethylene; PTFE, 0.2 μm, Hydrophobic, Advantec, Japan) and mixed with 300 μL of deionized water (for water packing). Samples were analyzed using a 1200 series HPLC system (Agilent Technologies, CA, USA) for qualitative and quantitative evaluation of pigments [27]. Pigments were separated using a Zobrax Eclipse XDB C8 column (4.6 × 250 mm, 5 μm, Agilent Technologies, CA, USA) with an injection volume of 100 μL. The mobile phases were (A) methanol: acetonitrile: aqueous pyridine solution (50: 25: 25, v/v/v) and (B) methanol: acetonitrile: acetone (60: 20: 60, v/v/v). The aqueous pyridine solution was mixed with 10 mL acetic acid and 20 mL pyridine, and then 900 mL of deionized water was added. Acetic acid was added dropwise to adjust the pH to 5.5. The final volume of the mixture was 1000 mL. The gradient condition of mobile phase A was increased slowly to 60% for 21 min, decreased to 5% for 6 min, maintained for 10 min, and increased slowly to 100% for 8 min. The flow rate of the mobile phase was 1.0 mL min$^{-1}$. The absorbance of each pigment was detected with UV-visible detector at 440 nm. Each peak was distinguished according to retention time and a spectrum of standards for Chlorophyll *a* (Chl. *a*; Sigma-Aldrich, ST Louis, MO, USA) and other pigments (Danish Hydraulic Institute; DHI, Hørsholm, Denmark). Pigment concentrations were calculated using the equation suggested by Park et al. [28], and were integrated from the surface to 1% light depth.

Pigment analysis using HPLC has the advantages of providing relatively reliable and consistent results in a short time. In addition, pico-sized phytoplankton and minor phytoplankton groups can be analyzed at the class level [29,30]. Cyanobacteria have very small cell sizes and aggregate to form colonies, making it time-consuming to enumerate cyanobacterial cells in the freshwater environment [31]. However, quantitative analysis of phytoplankton marker pigments has limitation in estimating the dominant phytoplankton. Mackey et al. [32] developed the CHEMTAX program to estimate the relative proportions of each phytoplankton composition using a steepest descent algorithm and factor analysis based on pigment/Chl. *a* ratios. The CHEMTAX program was developed for marine phytoplankton research, but has recently been applied to various freshwater environments [31,33–35]. The initial pigment/Chl. *a* ratios [33,34] used in this study was described in Table A1.

## 2.6. Statistical Analysis

Statistical tests were performed using IBM SPSS 23 (2015 SPSS Inc., IBM Corp., Armonk, NY, USA). Significant differences in parametric variables after log-transformation were determined by one-way analysis of variance (one-way ANOVA). Subsequently, Tukey's post-hoc test for

homogeneity of variances and Dunnett's T3 post-hoc test for heteroscedasticity of variances were performed to compare pairwise differences. The relative proportion of pigment-based phytoplankton composition calculated by the CHEMTAX program from four study sites (500 m upstream from weirs; ND-1, ND-4, ND-5, and ND-6) were compared to the microscopy results [36,37] by Pearson correlation analysis. The information of sampling dates for correlation analysis between CHEMTAX and cell counting are provided in Table A2. Principle component analysis (PCA) was performed using the statistical package R (R studio, version 1.1.463, Inc., Boston, MA, USA) to identify environmental factors relating primary production according to season using the data from the four study sites in front of the weir (ND-1, ND-4, ND-5, and ND-6). Spearman rank order correlation analysis was used to evaluate relationships among primary production and environmental factors, and to confirm correlation coefficients between the relative proportion of pigment-based phytoplankton composition calculated by the CHEMTAX program and environmental factors. For principal component analysis (PCA) and Spearman rank order correlation analysis, primary production (mg C m$^{-2}$ d$^{-1}$) and euphotic depth-integrated Chl. *a* concentrations (mg m$^{-2}$) were used and the other physicochemical variables were used by surface water data. Meanwhile, Chl. *a* concentrations (µg L$^{-1}$) were used for Spearman rank order correlation analysis between relative proportion of pigment-based phytoplankton composition and environmental factors. A *p*-value < 0.05 was considered to indicate statistical significance.

## 3. Results

### 3.1. Physicochemical Characteristics in the Regulated Water Environment

Heavy rainfall was concentrated between late August and early September (total, 376 mm), followed by the period from late June to early July (total, 249 mm) during our study period. This heavy rainfall induced an increase in discharge from all weirs (Figure 2). Water inflow during the study period ranged from 29 m$^3$ s$^{-1}$ to 1209 m$^3$ s$^{-1}$ and water discharge through the gate of the weirs ranged from 11.1 m$^3$ s$^{-1}$ to 1214 m$^3$ s$^{-1}$ at four study sites in front of the weirs (Table 1). The average water inflow and discharge at ND-6 were 274 ± 320 m$^3$ s$^{-1}$ and 276 ± 320 m$^3$ s$^{-1}$, respectively, showing the highest values among the study sites. The distance between ND-5 and ND-6 was the furthest (44 km) compared to the distance between other study sites. HRT ranged from 0 to 32 days, and the average value was lowest on September 6th (1 ± 0.6 days) after heavy rainfall and longest on August 8th (20 ± 9 days) at all study sites. Average LEC at all study sites showed the highest value on September 6th (2.8 ± 0.70 m$^{-1}$), followed by August 8th (2.7 ± 0.73 m$^{-1}$) (Table A3). Euphotic depth and secchi depth showed the lowest value on September 6th, followed by August 8th at all study sites. Water temperature ranged from 15.1 to 33.7 °C (Table 1, Figure 3a) and pH ranged from 7.3 to 10 (Table 1, Figure 3b).

$NH_4^+$, $NO_3^- + NO_2^-$, $PO_4^{3-}$, and $SiO_2$ concentrations were not significantly different among the study sites (*p* > 0.05). However, nutrient concentrations showed significant seasonal differences (*p* < 0.01) except for $NH_4^+$ concentrations (*p* > 0.05). $NH_4^+$ concentrations were low with a range of below the detection limit to 0.11 mg L$^{-1}$ at all study sites without noticeable seasonal variations (Figure 4a). $NO_3^- + NO_2^-$ concentrations varied widely from 0.57 to 4.5 mg L$^{-1}$ (Figure 4b). Concentrations of $NO_3^- + NO_2^-$ in summer were significantly different from those in spring (*p* < 0.05) and autumn (*p* < 0.01). The Average concentrations of $NO_3^- + NO_2^-$ at all study sites decreased gradually from April 27th (mean ± S.D. = 3.1 ± 0.70 mg L$^{-1}$) to August 8th (1.3 ± 0.69 mg L$^{-1}$) and then increased after August. $PO_4^{3-}$ concentrations at all study sites were generally low before September, ranging from below the detection limit to 0.19 mg L$^{-1}$ (Figure 4c). However, $PO_4^{3-}$ concentrations were significantly higher in autumn than in spring (*p* < 0.05) and summer (*p* < 0.01). Average concentrations of $PO_4^{3-}$ at all study sites were highest on September 6th (0.15 ± 0.035 mg L$^{-1}$) when heavy rainfall occurred. $SiO_2$ concentrations varied from below the detection limit to 10 mg L$^{-1}$ (Figure 4d). $SiO_2$ concentrations differed significantly among spring, summer, and autumn (*p* < 0.01). They decreased after April 27th, and then remained relatively low value until June 25th and then increased sharply after July.

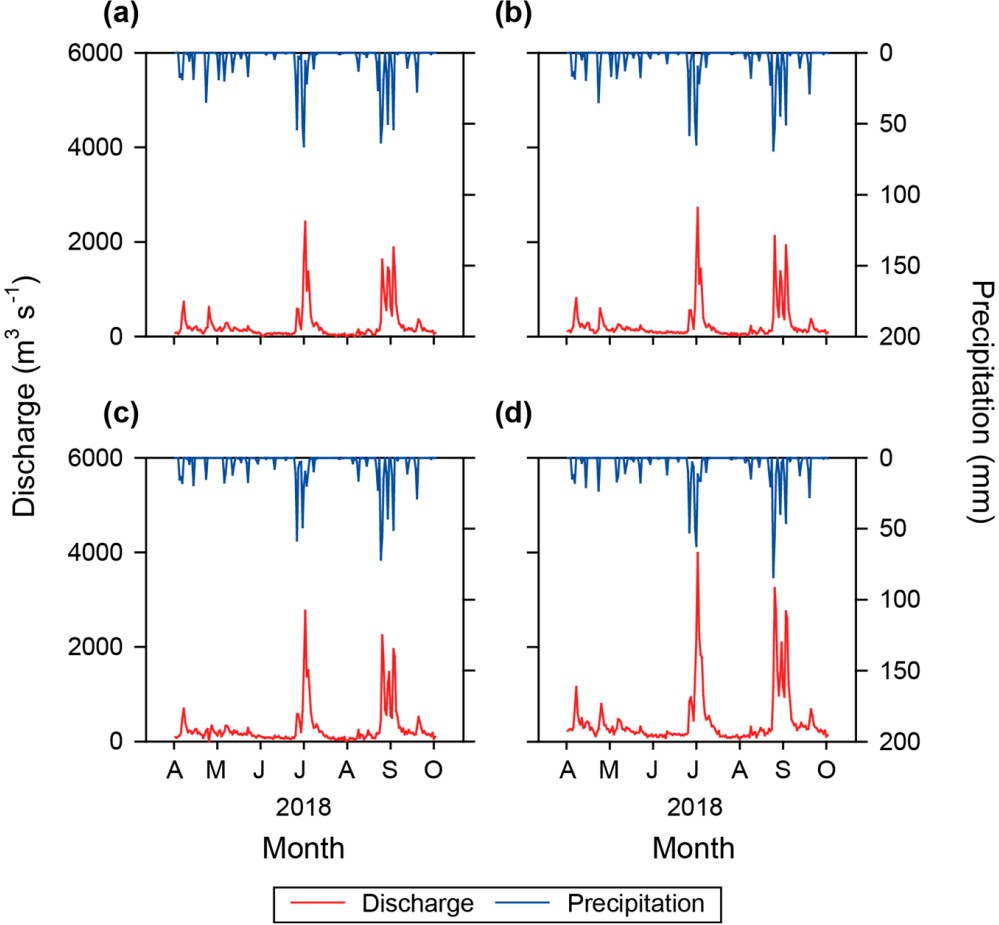

**Figure 2.** Discharge (m$^3$ s$^{-1}$) and precipitation (mm) from April to October 2018 at (**a**) Gangjeong-Goryeong weir; ND-1, (**b**) Dalseong weir; ND-4, (**c**) Hapcheon-Changnyeong weir; ND-5, and (**d**) Changnyeong-Haman weir; ND-6 (Water Environment Information System; http://water.nier.go.kr).

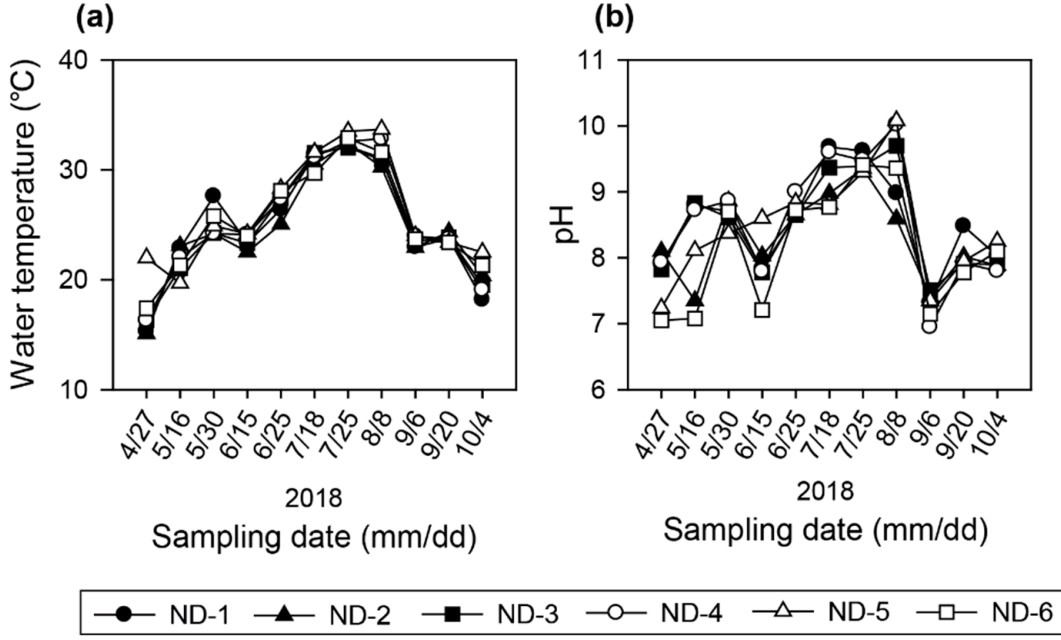

**Figure 3.** Temporal variations in (**a**) water temperature and (**b**) pH at each study site during the study period.

**Table 1.** Environmental parameters of the study sites in the Nakdong River (Water Environment Information System; http://water.nier.go.kr).

| Site No. | Study Sites | Distance between Study Sites | Water Level | Water Storage | Inflow | Discharge | Hydraulic Retention Time (HRT) | Water Temperature (WT) | pH |
|---|---|---|---|---|---|---|---|---|---|
| | | (km) | (EL.m) | ($\times 10^6$ m$^3$) | (m$^3$ s$^{-1}$) | (m$^3$ s$^{-1}$) | (days) | (°C) | |
| ND-1 | Gangjeong-Goryeong weir | – | 18.3<br>18.7<br>18.5 ± 0.19 | 73.5<br>80.1<br>77.0 ± 2.7 | 28.6<br>702<br>153 ± 196 | 11.6<br>686<br>147 ± 194 | 1.3<br>32<br>13 ± 9.8 | 15.3<br>32.1<br>25.0 ± 5.4 | 7.3<br>9.7<br>8.5 ± 0.7 |
| ND-2 | Samunjin bridge | ND-1–ND-2<br>(4) | –<br>–<br>– | –<br>–<br>– | –<br>–<br>– | –<br>–<br>– | –<br>–<br>– | 15.1<br>32.4<br>24.6 ± 5.0 | 7.4<br>9.4<br>8.3 ± 0.6 |
| ND-3 | Goryeong bridge | ND-2–ND-3<br>(12) | –<br>–<br>– | –<br>–<br>– | –<br>–<br>– | –<br>–<br>– | –<br>–<br>– | 16.3<br>32.0<br>24.9 ± 5.1 | 7.5<br>9.7<br>8.5 ± 0.7 |
| ND-4 | Dalseong weir | ND-3–ND-4<br>(3) | 13.6<br>13.9<br>13.7 ± 0.12 | 53.9<br>57.4<br>55.5 ± 1.3 | 29.2<br>580<br>151 ± 155 | 29.2<br>577<br>146 ± 154 | 0.43<br>19<br>6.9 ± 4.8 | 16.3<br>32.8<br>25.2 ± 5.4 | 7.0<br>10.0<br>8.6 ± 1 |
| ND-5 | Hapcheon-Changnyeong weir | ND-4–ND-5<br>(29) | 9.17<br>9.34<br>9.27 ± 0.059 | 54.5<br>56.0<br>55.4 ± 0.51 | 32.4<br>646<br>165 ± 179 | 41.5<br>649<br>170 ± 178 | 1.0<br>20<br>7.7 ± 6.1 | 19.7<br>33.7<br>26.2 ± 4.9 | 7.2<br>10.1<br>8.5 ± 0.8 |
| ND-6 | Changnyeong-Haman weir | ND-5–ND-6<br>(44) | 4.82<br>4.99<br>4.90 ± 0.059 | 97.3<br>101<br>98.8 ± 1.2 | 78.9<br>1209<br>274 ± 320 | 83.6<br>1214<br>276 ± 320 | 0.96<br>14<>6.9 ± 3.7 | 17.4<br>32.9<br>25.4 ± 4.8 | 7.1<br>9.4<br>8.1 ± 0.9 |

–: No information. EL.m: Elevation (m).

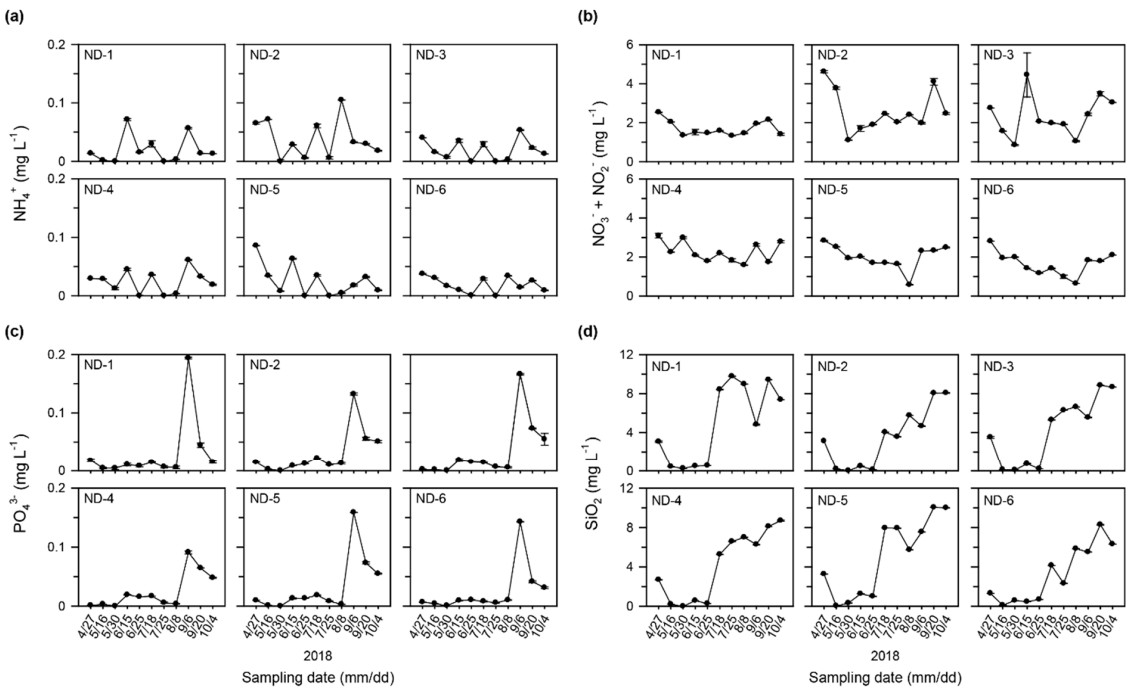

**Figure 4.** Spatio-temporal variations in concentrations of (**a**) $NH_4^+$, (**b**) $NO_3^- + NO_2^-$, (**c**) $PO_4^{3-}$, and (**d**) $SiO_2$ in the Nakdong River.

### 3.2. Seasonal and Spatial Variations in Primary Production

The primary productivity from surface to 1% light depth was measured from April to October 2018. During the study period, relative contributions of primary productivity were $70 \pm 12\%$ at the surface, $24 \pm 10\%$ at 50% light depth, $4.6 \pm 3.0\%$ at 12% light depth, and $1.1 \pm 0.91\%$ at 1% light depth. The water depth corresponding to 50% light depth ranged from 0.2 to 1.3 m water depth. This means that most primary productivity in the Nakdong River occurred within a water depth of 1 m. Depth-integrated primary productivity (primary production) varied widely with a range of $10–3252$ mg C m$^{-2}$ d$^{-1}$ ($534 \pm 919$ mg C m$^{-2}$ d$^{-1}$) at ND-1, $9–1850$ mg C m$^{-2}$ d$^{-1}$ ($568 \pm 539$ mg C m$^{-2}$ d$^{-1}$) at ND-2, $12–2839$ mg C m$^{-2}$ d$^{-1}$ ($887 \pm 890$ mg C m$^{-2}$ d$^{-1}$) at ND-3, $42–2545$ mg C m$^{-2}$ d$^{-1}$ ($898 \pm 792$ mg C m$^{-2}$ d$^{-1}$) at ND-4, $91–3519$ mg C m$^{-2}$ d$^{-1}$ ($1238 \pm 1217$ mg C m$^{-2}$ d$^{-1}$) at ND-5, and $91–3357$ mg C m$^{-2}$ d$^{-1}$ ($870 \pm 915$ mg C m$^{-2}$ d$^{-1}$) at ND-6 (Figure 5).

Average primary production at all study sites showed the highest value on April 27th ($2475 \pm 988$ mg C m$^{-2}$ d$^{-1}$), followed by August 8th ($1639 \pm 1045$ mg C m$^{-2}$ d$^{-1}$). Primary production showed the lowest value on September 6th at all study sites, with a range of $9–91$ mg C m$^{-2}$ d$^{-1}$. Primary production differed significantly depending on the season ($p < 0.01$) (Figure 6a). It was significantly high in spring ($p < 0.01$) and summer ($p < 0.01$) compared to autumn. There was spatial variation in primary production, except in autumn, which showed a significant seasonal difference (Figure 6b). Primary production was not significantly different among the study sites in spring and summer ($p > 0.05$).

### 3.3. Seasonal and Spatial Variations in Marker Pigment Concentrations of Phytoplankton

Phytoplankton have different marker pigments according to their classes. Concentrations of euphotic depth-integrated pigments from surface to 1% light depth were measured (Figure 7). Euphotic depth-integrated Chl. *a* concentrations were not significantly different according to study sites and seasons ($p > 0.05$). Average Chl. *a* concentration at all study sites was highest in spring ($44 \pm 29$ mg m$^{-2}$), followed by autumn ($33 \pm 25$ mg m$^{-2}$), and was relatively low in summer ($29 \pm 22$ mg m$^{-2}$).

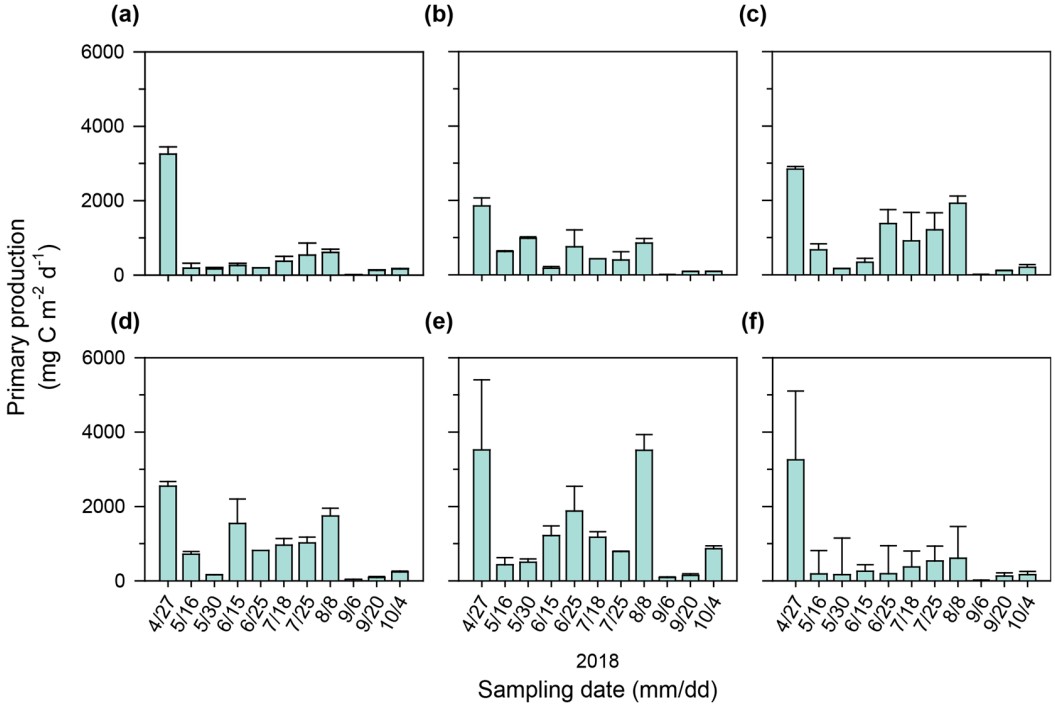

**Figure 5.** Euphotic depth-integrated primary productivity (primary production) at (**a**) Gangjeong-Goryeong weir; ND-1, (**b**) Samunjin bridge; ND-2, (**c**) Goryeong bridge; ND-3, (**d**) Dalseong weir; ND-4, (**e**) Hapcheon-Changnyeong weir; ND-5, and (**f**) Changnyeong-Haman weir; ND-6.

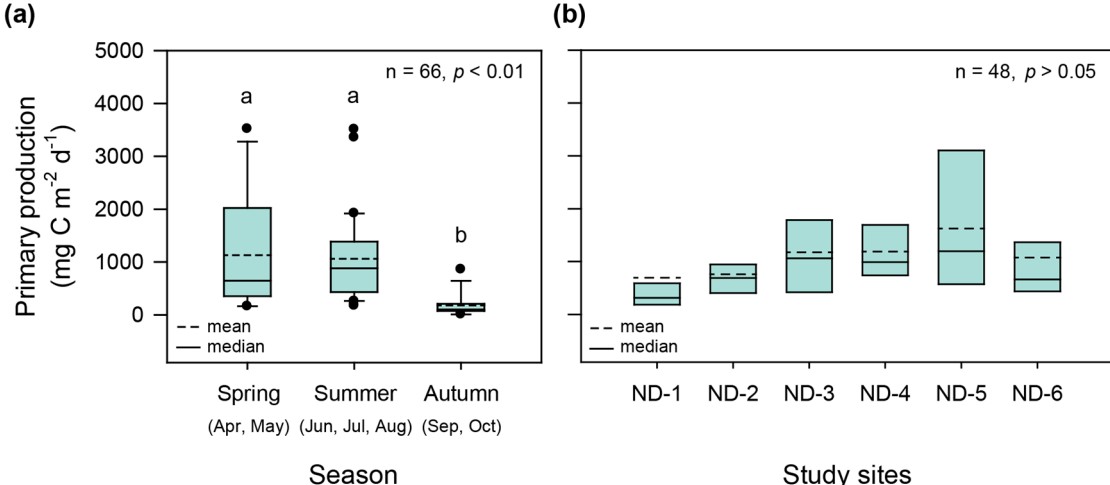

**Figure 6.** (**a**) Seasonal and (**b**) spatial variations in euphotic depth-integrated primary productivity (primary production) at all study sites in the Nakdong River. One-way analysis of variance (ANOVA) was performed to assess the significance of differences among seasons. The letters a and b in Figure 6a indicate significant difference by Tukey's post-hoc test. A box plot of Figure 6b was shown excluding water samples in autumn which showed a seasonal difference.

Seasonal variations in accessory pigment concentrations were similar at all study sites. Fucoxanthin (a marker pigment for diatoms) concentrations were high in spring, with an average value of $18 \pm 5.8$ mg m$^{-2}$ on April 27th, and $19 \pm 11$ mg m$^{-2}$ on May 16th, and decreased in summer. Alloxanthin (a marker pigment of cryptophytes) concentrations showed the highest value on September 20th ($4.5 \pm 1.7$ mg m$^{-2}$). The concentrations of Chl. *b*, lutein (marker pigments of chlorophytes) and zeaxanthin (a marker pigment of cyanobacteria) tended to increase in summer. Concentrations of Chl. *b* and lutein were highest on July 25th, with average values of $3.7 \pm 1.7$ mg m$^{-2}$ and $1.5 \pm 0.55$ mg m$^{-2}$,

respectively. Zeaxanthin concentrations were highest on July 18th ($1.3 \pm 0.62$ mg m$^{-2}$). Concentrations of Chl. *a* and accessory pigments decreased sharply on September 6th immediately after the heavy rainfall.

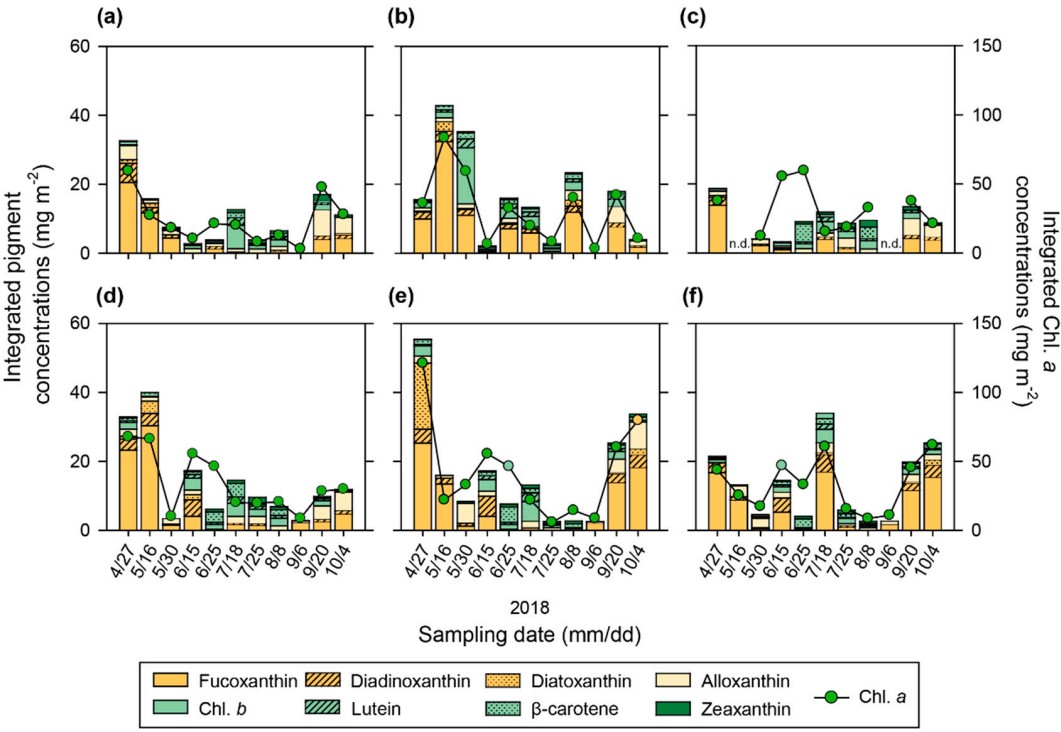

**Figure 7.** Euphotic depth-integrated Chl. *a* and accessory pigment concentrations (mg m$^{-2}$) at (**a**) Gangjeong-Goryeong weir; ND-1, (**b**) Samunjin bridge; ND-2, (**c**) Goryeong bridge; ND-3, (**d**) Dalseong weir; ND-4, (**e**) Hapcheon-Changnyeong weir; ND-5, and (**f**) Changnyeong-Haman weir; ND-6.

### 3.4. Relative Proportion of Pigment-Based Phytoplankton Composition Calculated by the CHEMTAX Program

The relative proportion of pigment-based phytoplankton composition calculated by the CHEMTAX program showed a good correlation with cell numbers measured by microscopy, except for cryptophytes (Figure 8). This finding indicated that the CHEMTAX program based on pigment analysis can accurately describe the relative proportion of phytoplankton composition in a freshwater environment.

The results calculated by the CHEMTAX program did not differ among the study sites ($p > 0.05$). Significant seasonal variations in relative proportion of pigment-based phytoplankton composition were observed in diatoms ($p < 0.01$), cryptophytes ($p < 0.05$), chlorophytes ($p < 0.01$), and cyanobacteria ($p < 0.01$). During the study period, *Aulacoseira* sp., *Cryptomonas* sp., *Eudorina* sp., and *Microcystis* sp. were dominant species in diatoms, cryptophytes, chlorophytes, and cyanobacteria, respectively [36,37]. The relative proportion of pigment-based diatoms ranged from 0 to 96% ($46 \pm 32$%), cryptophytes from 2 to 64% ($20 \pm 16$%), chlorophytes from 0 to 76% ($17 \pm 18$%), and cyanobacteria from 0 to 88% ($17 \pm 24$%) (Figure 9). The relative proportion of diatoms in summer was significantly different from those in spring and autumn ($p < 0.01$), but significant differences were not found between spring and autumn ($p > 0.05$). The relative proportion of diatoms was high from April 27th ($89 \pm 6.4$%) to June 15th ($52 \pm 23$%). It decreased to $5.5 \pm 13$% on August 8th, and they dominated again on September 6th, with a value of $85 \pm 9.3$%. The relative proportion of pigment-based cryptophytes was high on May 30th ($41 \pm 24$%), followed by September 20th ($36 \pm 24$%). It was significantly different in summer and autumn ($p < 0.01$). The relative proportion of pigment-based chlorophytes and cyanobacteria in summer were significantly different from those in spring and autumn ($p < 0.01$, respectively). The proportion of chlorophytes was low on April 27th ($3.4 \pm 5.5$%) and started to increase in summer. It was the highest on July 18th ($49 \pm 24$%) and then decreased after summer. Seasonal variations in cyanobacteria proportion were similar to those observed for chlorophytes. Cyanobacteria were not measured from April 27th to May

16th, were detected for the first time on May 30th (5.9 ± 6.1%), and largely dominated on August 8th (61 ± 20%).

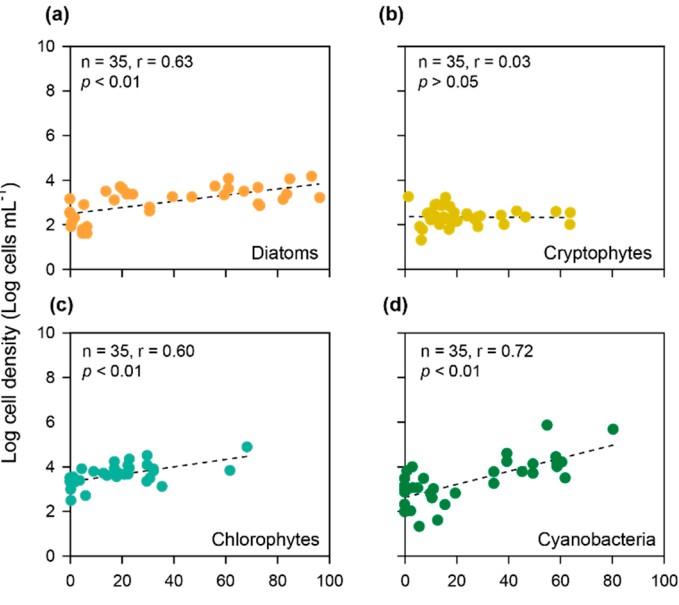

**Figure 8.** Relationships between relative proportion of pigment-based phytoplankton composition (%) calculated by the CHEMTAX program and Log cell density (Log cells mL$^{-1}$) for (**a**) diatoms, (**b**) cryptophytes, (**c**) chlorophytes, and (**d**) cyanobacteria.

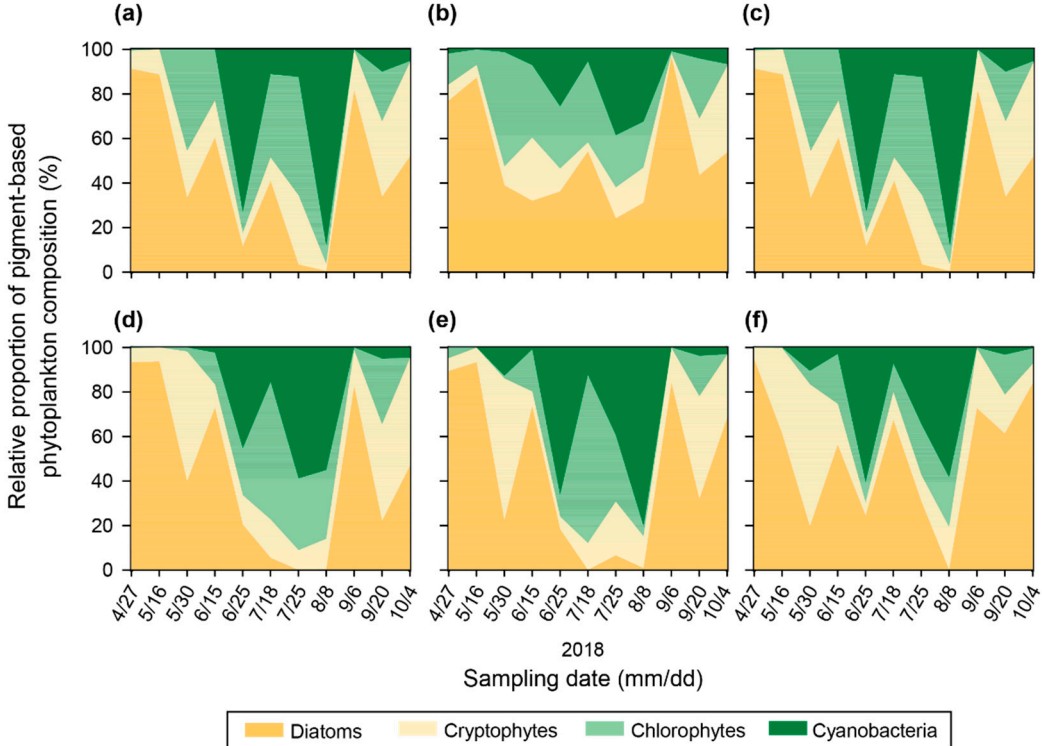

**Figure 9.** Relative proportion of pigment-based phytoplankton composition (%) calculated by the CHEMTAX program at (**a**) Gangjeong-Goryeong weir; ND-1, (**b**) Samunjin bridge; ND-2, (**c**) Goryeong bridge; ND-3, (**d**) Dalseong weir; ND-4, (**e**) Hapcheon-Changnyeong weir; ND-5, and (**f**) Changnyeong-Haman weir; ND-6.

## 4. Discussion

### 4.1. Seasonal and Spatial Variations in Primary Production in a Continuous Weir System

Wetzel [38] classified primary production above 1000 mg C m$^{-2}$ d$^{-1}$ as eutrophic status. In this study, primary production was once higher than 1000 mg C m$^{-2}$ d$^{-1}$ at the upper study sites (ND-1 and ND2) on April 27th. In comparison, higher primary production over 1000 mg C m$^{-2}$ d$^{-1}$ were detected four times at the middle study sites (ND-3 and ND-4), five times at ND-5, and three times at the lower study site (ND-6) (Figure 5). These results indicate that primary production increased from the upper to lower sites of the continuous weir system in the Nakdong River, with the exception of ND-6. In addition, mean and median primary production tended to increase gradually from ND-1 to ND-5 (Figure 6b). Since ND-6 is located at the furthest site (44 km) from ND-5, environment conditions could be different at ND-6 which could be influenced by much larger inflow of the water from local tributaries compared to other study sites (Table 1).

The Korean Ministry of Environment has implemented water level management (opening of weirs) to improve water quality for several weirs, including all study sites, since June 2017. River discharge directly affects hydrological factors linked to water fluxes such as turbidity, nutrients loads, and phytoplankton biomass [39]. Opening weirs not only controls phytoplankton cell density and primary production, but also reduces or prevents the proliferation of algal blooms in the regulated water environment [2,17]. Water level management was conducted at all study sites during the study period. Model results at Gangjeong-Goryeong weir in the Nakdong River demonstrated that Chl. *a* concentrations and cell density decreased considerably by lowering water level [40] because phytoplankton flushed out downstream. Flushing to reduce algal blooms, however, is only effective under low nutrient conditions [41]. During the study period, phytoplankton seemed to be incubated under the sufficient light and nutrients conditions through downstream transport in a continuous weir system.

### 4.2. Key Factors Controlling Primary Production in a Continuous Weir System

To investigate the historical changes in primary production in the Nakdong River before and after weir construction, we compared our findings with previously published data for the Nakdong River (Table A4). The range of primary production decreased considerably after the construction of weirs compared to those before the construction of weirs, although the methodology used to assess primary productivity was slightly different ($^{14}$C tracer versus the $^{13}$C tracer used in the current study). There was no significant difference in air temperature before and after weir construction, but the water temperature increased by approximately 5% due to the increasing HRT in the Nakdong River after the construction of weirs [21]. Meanwhile, total phosphorus (TP) concentrations decreased noticeably by 47 to 72% due to reinforcement of regulations regarding waste water treatment discharge into the Nakdong River [21]. Primary production decreases when TP availability is limited by nutrient management [7]. Water temperature was a major factor that affects primary productivity in high TP conditions, but the effect of water temperature was found to decrease at low TP concentrations in Lake Geneva, Switzerland [42]. Additionally, water temperature was found to have little effect on phytoplankton under low nutrient conditions in Lake Bassenthwaite, England [10]. These results suggest that the reduction in TP concentrations through the regulation of wastewater discharge after the construction of weirs in the Nakdong River effectively decreased primary production.

Principal component analysis (PCA) was performed to identify the relationships among environmental factors (Figure 10). The two axes explained a cumulative variance of 53.95%. Although primary production in the Nakdong River did not differ between spring and summer, PCA showed clear seasonal differences in physicochemical environmental factors affecting primary production. For the first principal component (PC1, explaining 32.67% of the variance), Chl. *a* concentrations, N/P ratio, and dissolved inorganic nitrogen (DIN) such as NH$_4^+$ and NO$_3^-$ + NO$_2^-$ were positively loaded in spring and autumn. By contrast, light intensity, water temperature,

HRT, pH, POC, and PN were positively loaded in summer. On the second principal component (PC2, explaining 21.28% of the variance), $PO_4^{3-}$ was positively loaded in autumn, but negatively loaded in spring and summer. Primary production was negatively correlated with $PO_4^{3-}$ ($r = -0.41$, $p < 0.01$) (Table A5) due to large $PO_4^{3-}$ utilization by phytoplankton when primary productivity was high in spring and summer. As a result, primary production was positively correlated with the N/P ratio ($r = 0.44$, $p < 0.01$). The N/P ratio showed a strong negative correlation with $PO_4^{3-}$ ($r = -0.94$, $p < 0.01$). It seems that the N/P ratio was mainly regulated by $PO_4^{3-}$ concentrations.

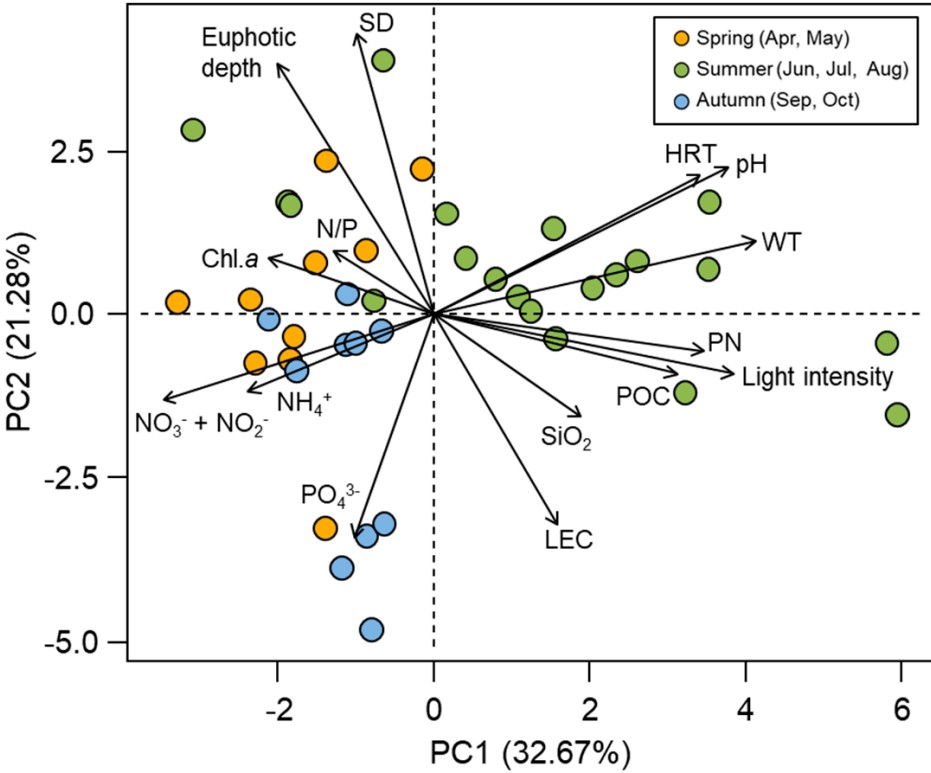

**Figure 10.** Principal component analysis (PCA) to determine relationships among environmental factors (Light intensity ($\mu$mol m$^{-2}$ s$^{-1}$), LEC; Light extinction coefficient (m$^{-1}$), Euphotic depth (m), SD; Secchi depth (m), WT.; Water temperature (°C), pH, HRT (days), $NH_4^+$ (mg L$^{-1}$), $NO_3^- + NO_2^-$ (mg L$^{-1}$), $PO_4^{3-}$ (mg L$^{-1}$), $SiO_2$ (mg L$^{-1}$), N/P, POC (mg L$^{-1}$), PN (mg L$^{-1}$), Chl. *a* (mg m$^{-2}$)).

The $PO_4^{3-}$ concentration increased sharply by 10-fold after heavy rainfall on September 6th. A large amount of TP inputs from non-point source pollutions was previously reported in the Nakdong River after rainfall [43]. Approximately 60% of the TP load in the Nakdong River could be originated from non-point source pollutions [21]. The water condition of this study sites was the $PO_4^{3-}$ limited environment (Figure 11) based on the criteria suggested by Justić et al. [44]. The inflow of $PO_4^{3-}$ can promote primary productivity in aquatic environments where phosphate is a limiting factor [9,34,45]. Furthermore, excess $PO_4^{3-}$ concentrations and increasing N/P ratio can induce severe cyanobacterial blooms [4]. However, rainfall can also cause high turbidity and low light availability as a result of sediment resuspension, which inhibits photosynthesis and algal growth [8]. Heavy rainfall increased discharge, which is why the average HRT on September 6th was approximately 1 day at all study sites (Table 1). In addition, euphotic depth and secchi depth were shallow due to turbidity at that time (Table A3). Despite the high $PO_4^{3-}$ input right after heavy rainfall, a short HRT and light limitations due to turbidity can cause a low primary production [5]. For this reason, the lower primary productivity was measured in autumn than the other seasons, even though a large amount of $PO_4^{3-}$ was introduced by rainfall.

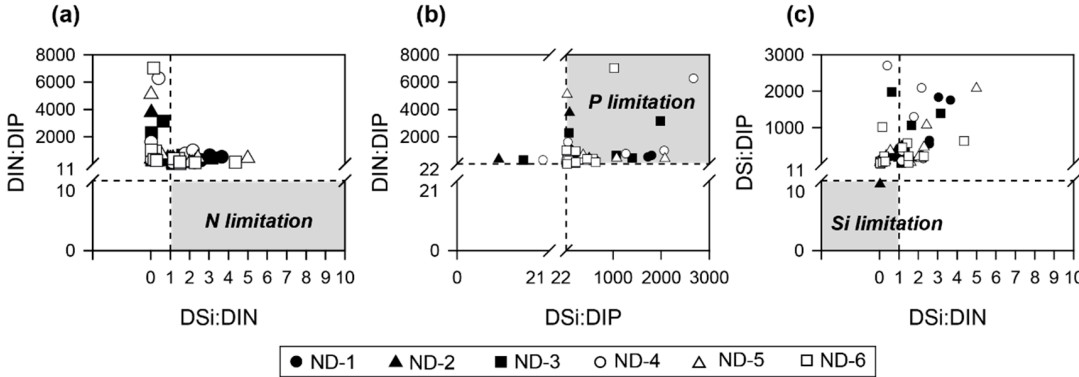

**Figure 11.** Assessment of (**a**) nitrogen limitation, (**b**) phosphate limitation, and (**c**) silicate limitation at all study sites using the criteria of nutrient ratios suggested by Justic et al. [44].

### 4.3. Favorable Environmental Conditions for cHABs in a Continuous Weir System

Cyanobacteria were predominant in summer, while diatoms dominated in spring and autumn according to the CHEMTAX analysis (Figure 5). The relative proportion of pigment-based diatoms showed a negative relationship with $SiO_2$ concentrations (r = −0.32, p < 0.01) (Table 2), likely because $SiO_2$ is required by diatom cells for diatom blooms in spring [46]. The relative proportion of pigment-based cryptophytes was relatively higher in September and October compared to other seasons. Cryptophytes grow well under low light conditions caused by moderate turbulence [47,48]. Mixing of water and the decline in light intensity in autumn have provided a favorable environmental condition for the growth of cryptophytes.

**Table 2.** Spearman rank order correlation coefficients between environmental factors and the relative proportion of pigment-based phytoplankton composition calculated by the CHEMTAX program.

| | Light | SD | HRT | WT | pH | $NH_4^+$ | $NO_3^- + NO_2^-$ | $PO_4^{3-}$ | $SiO_2$ | Chl. *a* |
|---|---|---|---|---|---|---|---|---|---|---|
| Diatoms | −0.223 | −0.231 | −0.732 ** | −0.779 ** | −0.698 ** | 0.357 * | 0.617 ** | 0.146 | −0.322 * | 0.338 * |
| Cryptophytes | −0.349 * | 0.157 | 0.162 | 0.003 | −0.053 | −0.134 | −0.071 | 0.028 | 0.141 | −0.382 * |
| Chlorophytes | 0.253 | 0.196 | 0.509 ** | 0.653 ** | 0.466 ** | 0.000 | −0.446 ** | 0.135 | 0.379 * | −0.019 |
| Cyanobacteria | 0.275 | 0.201 | 0.762 ** | 0.794 ** | 0.713 ** | −0.494 ** | −0.752 ** | −0.134 | 0.247 | −0.246 |

* p < 0.05, ** p < 0.01.

In this study, *Microcystis* sp. was the dominant genera when cHABs occurred in the Nakdong River. The Korean alert system of algal blooms, as criteria for cyanobacteria, have three levels which are Caution (1000 cells $mL^{-1}$), Warning (10,000 cells $mL^{-1}$), and Outbreak (1,000,000 cells $mL^{-1}$). The exceeding 5000 cells $mL^{-1}$ of cyanobacteria occurred 62% of all the study sites in summer (from June to August), and the highest cyanobacterial cell density was 1,264,052 cells $mL^{-1}$ at Hapcheon-Changnyeong weir in August 2018 [49]. In addition, the average cyanobacteria cell density was 694,667 cells $mL^{-1}$ at the Dalseong weir, followed by 453,283 cells $mL^{-1}$ at the Hapcheon-Changnyeong weir in summer [37]. This genus is known to produce toxins such as MCs [50]. Their large colony sizes and toxicity can prevent the consumption by zooplankton, and their buoyancy control allows them to adapt rapidly and migrate vertically to thermally stable layers [51]. MCs can be transported to upper trophic levels through aquatic food web [52] and have an adverse effect on coastal habitats when they are discharged into estuaries [16,53].

Water temperature is one of the most important factors affecting algal growth, phytoplankton diversity, and interspecific competition [41]. Water temperature showed a significant negative correlation with diatoms (r = −0.78, p < 0.01) while it was positively correlated with chlorophytes (r = 0.65, p < 0.01) and cyanobacteria (r = 0.79, p < 0.01) (Table 2). The average water temperature at all study sites was 21 ± 3.8 °C in spring, 29 ± 3.5 °C in summer, and 22 ± 1.9 °C in autumn. Diatoms abundance was high at low water temperatures (5–20 °C) [54]. The optimum water temperatures for cyanobacteria

and chlorophytes range from 25 to 35 °C and from 28 to 35 °C, respectively, and increasing water temperature tends to induce more abundant cyanobacteria than chlorophytes in a lab experiment [3]. The relative proportion of cyanobacteria (mainly *Microcystis* sp.) was positively correlated with HRT (r = 0.76, $p < 0.01$) as well as water temperature in this study. The long HRT caused an increase in water temperature due to stagnant water flow [21]. In addition, cyanobacteria have several advantages in stratified conditions due to their ability to form colonies or filaments and to use their buoyancy to migrate vertically compared to chlorophytes [3,51]. During the study period, the increase in HRT caused by continuous weirs may have decreased physical stresses, including turbulence and flow velocity, to cyanobacteria at all study sites. The increased HRT could have provided a longer incubation time (growing period) for cyanobacterial cells, explaining the enhanced magnitude of cHABs in summer (from late July to mid-August). The longer HRT resulted from low volumes of inflow water was strongly related to the small amount of rainfall compared to other periods. On the contrary, the relative proportion of pigment-based cyanobacteria was low from late June to early July due to a short HRT resulting from frequent rainfall. Complex hydrological factors such as water temperature and HRT affected phytoplankton community composition more than nutrient concentrations in Lake Poyang, China [6] and also the Nakdong River [13]. These environmental conditions could be potential reasons for the substantial increase of harmful algae (mainly *Microcystis* sp.) cell density with a range of 57–1264,052 (97,506 ± 206,321) cells mL$^{-1}$ during summer (June to August) in 2018 compared to that right after the construction of weirs (range of 0–50,832 (8312 ± 9829) cells mL$^{-1}$) in 2013 [49]. Therefore, harmful cyanobacteria proliferated intensively in water environment with elevated temperatures and the increased HRT conditions created by continuous weirs in the Nakdong River.

## 5. Conclusions

We investigated primary productivity and the relative proportion of pigment-based phytoplankton composition in the Nakdong River where cHABs occur every year. Primary production showed a significant negative correlation with $PO_4^{3-}$ due to large $PO_4^{3-}$ utilization, demonstrating phosphate-limited condition in the Nakdong River. The relative proportion of pigment-based harmful algae (*Microcystis* sp.) increased due to the enhanced HRT and elevated water temperature, resulting in the occurrence of cHABs in a continuous weir system in summer. A short HRT as well as limited light availability caused by high turbidity inhibited phytoplankton primary production despite the high concentrations of $PO_4^{3-}$ as a result of heavy rainfall inflow in September 6th. Therefore, to reduce HRT through water discharge could be one of solutions for controlling primary production including cHABs in a continuous weir system. This study provides useful information regarding to the key factors controlling primary production and occurrence of cHABs in a continuous weir system.

**Author Contributions:** Conceptualization, J.C., S.H.L., K.C., and K.-H.S.; methodology, J.C., J.O.M., B.C., J.J.K., H.L.; formal analysis, J.C. and J.J.K.; investigation, J.C., J.O.M., B.C., and J.J.K.; writing—original draft preparation, J.C., J.O.M., and D.K.; writing—review and editing, S.H.L., J.J., and K.-H.S.; visualization, J.C.; supervision, K.-H.S.; project administration, K.C.; funding acquisition, S.H.L. and K.-H.S. All authors have read and agreed to the published version of the manuscript.

**Funding:** This research was funded by the K-Water Institute (project: Analysis of phytoplankton primary productivity and utilization rate of nutrients) and National Research Foundation of Korea (NRF) (No. NRF-2016R1E1A1A01943004).

**Acknowledgments:** We would like to thank Hyuntae Choi, Seung-Hee Kim, Myungjoon Kim, and Naeun Jo in Hanyang University and Pusan National University for their assistance during fieldwork. This work was supported by a K-water Institute grant (project: Analysis of phytoplankton primary productivity and utilization rate of nutrients) and a National Research Foundation of Korea (NRF) grant funded by the Korean government (Ministry of Science and ICT) (No. NRF-2016R1E1A1A01943004).

**Conflicts of Interest:** The authors declare no conflicts of interest.

## Appendix A

**Table A1.** Initial pigment/Chl. *a* ratios suggested by Schlüter et al. [33] and Paerl et al. [34].

|  | Fuco | Neo | Viol | Diad | Anth | Myx | Allo | Lut | Zea | Chl. *b* | ß-car | Ech |
|---|---|---|---|---|---|---|---|---|---|---|---|---|
| Diatoms | 0.51 | 0 | 0 | 0.074 | 0 | 0 | 0 | 0 | 0 | 0 | 0.003 | 0 |
| Cryptophytes | 0 | 0 | 0 | 0 | 0 | 0 | 0.37 | 0 | 0 | 0 | 0.001 | 0 |
| Chlorophytes | 0 | 0.038 | 0.026 | 0 | 0.016 | 0 | 0 | 0.15 | 0 | 0.36 | 0.003 | 0 |
| Cyanobacteria | 0 | 0 | 0 | 0 | 0 | 0.14 | 0 | 0 | 0.28 | 0 | 0.097 | 0.076 |

Fuco; Fucoxanthin, Neo; Neoxanthin, Viol; Violaxanthin, Diad; Diadinoxanthin, Anth; Antheraxanthin, Allo; Alloxanthin, Lut; Lutein, Zea; Zeaxanthin, Chl. b; Chlorophyll *b*, β-car; β carotene, Ech; Echinenon.

## Appendix B

**Table A2.** Sampling dates information for Pearson correlation analysis between CHEMTAX and cell counting.

| Method | Sampling Date (mm/dd) in 2018 | | | | | | | | | | | | |
|---|---|---|---|---|---|---|---|---|---|---|---|---|---|
| CHEMTAX | 4/27 | 5/16 | 5/30 | 6/15 | 6/25 | 7/18 | 7/25 | 4/25 | 8/8 | 8/8 | 9/6 | 9/20 | 10/4 |
| Cell counting | 4/18 | 5/16 | 5/30 | 6/11 | 6/20 | 7/19 | 7/24 | 7/27 | 8/1 | 8/7 | 9/11 | 9/17 | 10/2 |

## Appendix C

**Table A3.** Light extinction coefficients, euphotic depth, and secchi depth in the Nakdong River.

| | Study Sites | | | | | | | | | | | | | | | | | |
| | Gangjeong-Goryeong Weir | | | Samunjin Bridge | | | Goryeong Bridge | | | Dalseong Weir | | | Hapcheon-Changnyeong Weir | | | Changnyeong-Haman Weir | | |
| Date | LEC | $Z_{eu}$ | SD | LEC | $Z_{eu}$ | SD | LEC | $Z_{eu}$ | SD | LEC | $Z_{eu}$ | SD | LEC | $Z_{eu}$ | SD | LEC | $Z_{eu}$ | SD |
| (mm/dd) | $(m^{-1})$ | (m) | (m) | $(m^{-1})$ | (m) | (m) | $(m^{-1})$ | (m) | (m) | $(m^{-1})$ | (m) | (m) | $(m^{-1})$ | (m) | (m) | $(m^{-1})$ | (m) | (m) |
|---|---|---|---|---|---|---|---|---|---|---|---|---|---|---|---|---|---|---|
| 4/27 | 1.53 | 3.5 | 0.9 | 1.18 | 4.0 | 1.0 | 1.65 | 2.3 | 1.2 | 1.51 | 4.0 | 1.2 | 1.61 | 3.5 | 1.6 | 5.37 | 2.0 | 0.7 |
| 5/16 | 1.50 | 3.5 | 1.6 | 1.56 | 3.0 | 1.1 | – | 3.5 | – | 1.03 | 4.0 | 1.2 | 1.28 | 3.5 | 1.0 | 1.29 | 3.0 | 1.0 |
| 5/30 | 1.02 | 4.5 | 2.0 | 3.23 | 3.0 | 0.9 | 0.96 | 5.0 | 2.1 | 0.81 | 6.0 | 2.7 | 0.69 | 8.0 | 3.0 | 1.20 | 5.0 | 2.0 |
| 6/15 | 1.30 | 4.5 | 2.5 | 2.30 | 2.0 | 1.5 | 0.94 | 3.6 | 2.5 | 0.93 | 6.0 | 3.1 | 0.81 | 5.0 | 2.0 | 0.93 | 3.0 | 1.0 |
| 6/25 | 0.94 | 8.0 | 2.9 | 1.05 | 3.5 | 1.5 | 0.88 | 6.5 | 1.9 | 1.47 | 4.5 | 1.9 | 1.45 | 3.0 | 1.0 | 1.47 | 3.0 | 1.4 |
| 7/18 | 1.32 | 4.0 | 2.0 | 1.49 | 3.5 | 1.3 | 1.02 | 4.5 | 1.7 | 1.30 | 4.0 | 1.4 | 1.30 | 4.0 | 1.2 | 1.81 | 2.5 | 1.2 |
| 7/25 | 1.65 | 4.0 | 1.8 | 4.12 | 1.9 | 0.9 | 1.59 | 3.3 | 1.0 | 1.55 | 3.5 | 1.0 | 1.18 | 3.0 | 1.2 | 2.99 | 3.0 | 1.2 |
| 8/8 | 1.76 | 2.5 | 1.4 | 1.82 | 3.0 | 1.3 | 3.53 | 2.0 | 0.3 | 3.23 | 1.5 | 0.4 | 2.75 | 2.0 | 0.8 | 2.95 | 2.0 | 0.5 |
| 9/6 | 3.18 | 1.3 | 0.1 | 3.63 | 1.4 | 0.1 | – | 1.5 | – | 2.80 | 1.6 | 0.3 | 1.91 | 2.5 | 0.2 | 2.23 | 2.5 | 0.2 |
| 9/20 | 1.11 | 4.5 | 1.3 | 1.38 | 4.0 | 1.4 | 1.30 | 4.5 | 1.3 | 1.29 | 4.0 | 1.1 | 1.35 | 4.0 | 1.0 | 1.54 | 2.5 | 0.7 |
| 10/4 | 1.40 | 3.5 | 0.8 | 1.63 | 2.0 | 0.8 | 1.11 | 4.0 | 1.0 | 0.79 | 5.5 | 1.1 | 1.18 | 4.0 | 1.0 | 1.53 | 3.0 | 1.0 |

–: not determined. We could not get the data because we could not work on board. LEC: Light extinction coefficient. $Z_{eu}$: Euphotic depth. SD: Secchi depth.

## Appendix D

**Table A4.** Previous studies of primary production in the Nakdong River.

| Region | Methods | Sampling Date | Primary Production (mg C m$^{-2}$ d$^{-1}$) | References |
|---|---|---|---|---|
| Ganjeong weir–Haman weir | $^{13}$C | 2017.10–2017.12 | 61–933 | [55] |
| Nakdong River estuary | $^{14}$C | 1991.10–1992.09 | 543–4112 | [56] |
| Gangjeong–Hagu | $^{14}$C | 1994.07–1994.08 | 8000–23,000 | [57] |
| Seonakdong River | $^{14}$C | 1996.01–1996.10 | 2000–181,000 | [58] |
| Ganjeong weir–Haman weir | $^{13}$C | 2018.04–2018.10 | 8.9–3519 | This study |

## Appendix E

**Table A5.** Spearman rank order correlation coefficients between environmental factors and primary production.

| | Light | LEC | $Z_{eu}$ | SD | HRT | WT | pH | $NH_4^+$ | $NO_3^- + NO_2^-$ | $PO_4^{3-}$ | $SiO_2$ | POC | PN | N/P | Chl. $a$ | P.P. |
|---|---|---|---|---|---|---|---|---|---|---|---|---|---|---|---|---|
| Light | 1.000 | 0.620 ** | −0.486 ** | −0.220 | 0.303 | 0.694 ** | 0.599 ** | −0.377 * | −0.400 * | −0.141 | 0.395 * | 0.455 ** | 0.379 * | 0.027 | −0.508 ** | 0.200 |
| LEC | | 1.000 | −0.784 ** | −0.566 ** | −0.134 | 0.156 | 0.106 | −0.081 | −0.106 | 0.008 | 0.256 | 0.467 ** | 0.290 | −0.089 | −0.290 | 0.092 |
| $Z_{eu}$ | | | 1.000 | 0.771 ** | 0.142 | −0.131 | 0.033 | 0.092 | 0.162 | 0.007 | −0.150 | −0.510 ** | −0.332 * | 0.100 | 0.341 * | −0.075 |
| SD | | | | 1.000 | 0.379 * | 0.185 | 0.269 | −0.026 | −0.141 | −0.250 | −0.257 | −0.302 | −0.163 | 0.323 * | 0.213 | 0.099 |
| HRT | | | | | 1.000 | 0.641 ** | 0.672 ** | −0.523 ** | −0.737 ** | −0.383 * | 0.157 | 0.166 | 0.237 | 0.223 | −0.155 | 0.163 |
| WT | | | | | | 1.000 | 0.778 ** | −0.410 ** | −0.679 ** | −0.189 | 0.249 | 0.331 * | 0.397 * | 0.026 | −0.409 ** | 0.243 |
| pH | | | | | | | 1.000 | −0.525 ** | −0.555 ** | −0.410 ** | 0.197 | 0.361 * | 0.468 ** | 0.259 | −0.201 | 0.326 * |
| $NH_4^+$ | | | | | | | | 1.000 | 0.481 ** | 0.311 | −0.050 | −0.159 | −0.160 | −0.200 | −0.059 | −0.051 |
| $NO_3^- + NO_2^-$ | | | | | | | | | 1.000 | 0.242 | −0.032 | −0.152 | −0.232 | 0.015 | 0.380 * | −0.049 |
| $PO_4^{3-}$ | | | | | | | | | | 1.000 | 0.442 ** | −0.245 | −0.350 * | −0.938 ** | −0.076 | −0.408 ** |
| $SiO_2$ | | | | | | | | | | | 1.000 | 0.141 | 0.108 | −0.496 ** | −0.262 | −0.175 |
| POC | | | | | | | | | | | | 1.000 | 0.922 ** | 0.223 | 0.028 | 0.627 ** |
| PN | | | | | | | | | | | | | 1.000 | 0.308 | 0.085 | 0.709 ** |
| N/P | | | | | | | | | | | | | | 1.000 | 0.199 | 0.440 ** |
| Chl. $a$ | | | | | | | | | | | | | | | 1.000 | 0.304 |
| P.P. | | | | | | | | | | | | | | | | 1.000 |

Light; Light intensity, LEC; Light extinction coefficient, $Z_{eu}$; Euphotic depth, SD; Secchi depth, HRT; Hydraulic retention time, WT; Water temperature, POC; Particulate organic carbon, PN; Particulate nitrogen, P.P.; Primary production. * $p < 0.05$, ** $p < 0.01$.

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
