# Peer review of "Key Factors Controlling Primary Production and Cyanobacterial Harmful Algal Blooms (cHABs) in a Continuous Weir System in the Nakdong River, Korea"

_sustainability, doi:10.3390/su12156224_

Round 1

Reviewer 1 Report

General Comments: Manuscript is well written and the topic is interesting to the readres due to its significance of contents. However, primary productivity results can be improved, as well as its discussion. See sugestions in specific comments.

Specific comments:

Abstract- Well written and clear.

Introduction- lines 48 to 58- Suggestion- insert this information in the methodology section.

Methods

Lines 130 to 131

How do the authors estimate the light attenuation to collect water samples in 50% to 1% light depth to perform in situ incubations?

Please inform the total PAR and the attenuation for each incubation in the manuscript. I suggest a table with these data.

I suggest to rewrite the primary productivity methodology including this information.

Lines 164-165- please inform the methodology to qualitative and quantitative phytoplankton analyses under microscopy.

Results:

Lines 2-20- I suggest to include PAR data here, as well as vertical variation in PP. PE curves can be added here, and parameters such as alfa and Pmax calculated.

Suggestion: Present the PCA results in this section.

Discussion:

Line 65- Primary production was high in spring and summer (Figure 2a)- This sentence must be in the Results section.

First paragraph- lines 65 to 75- It is difficult to compare PP values without PAR values. I suggest to include PAR variation, or euphotic zone depth variation Zeu (Secchi) . Higher values during summer and spring can be related to light availability.

Lines 106-120- Please, insert PCA results in the Result section.

Suggestion- If PAR data or  Zeu depth data are available, including at least one of these variables to the PCA can be interesting, specially considering the periods with high cryptophytes densities.

Conclusion- Very clear and well written.

Author Response

We appreciate for the Reviewer 1 comments.

We completed the answers to the comments and suggestions of Reviewer 1, and revised the manuscript accordingly.

In addition, we wrote the “Authorship change form” due to added one author.

Response to Reviewer 1 Comments

Reviewer #1

General points:

Manuscript is well written and the topic is interesting to the readers due to its significance of contents. However, primary productivity results can be improved, as well as its discussion. See suggestions in specific points.

Response: The authors greatly preciate the Reviewer #1 for the positive points. The manuscript has been revised carefully as the Reviewer #1’s suggestions and points. Thanks a lot. Number of reference, figure, and table were changed while revising the manuscript. So, Figure number and table number in point may be different from those in response. Please also check the newly attached Figures and Tables.

Specific points:

Abstract

Well written and clear.

Introduction

Point 1: lines 48 to 58- Suggestion- insert this information in the methodology section.

Response 1: We agreed with Reviewer #1’s point, so information about the CHEMTAX program was transferred from introduction to methodology section (Line: 154166).

Revision 1:

“Pigment analysis using HPLC has the advantages of providing relatively reliable and consistent results in a short time. In addition, pico-sized phytoplankton and minor phytoplankton groups can be analyzed at the class level [28, 29]. Cyanobacteria have very small cell sizes and aggregate to form colonies, making it time-consuming to enumerate cyanobacterial cells in the freshwater environment [30]. However, quantitative and qualitative measurements of phytoplankton marker pigments have limitations in estimating the phytoplankton community and distribution. Mackey et al. [31] developed the CHEMTAX program to assess the contribution of each community to phytoplankton biomass using a steepest descent algorithm and factor analysis based on pigment/Chl. a ratios. The CHEMTAX program was developed for marine phytoplankton research, but has recently been applied to various freshwater environments [30, 32–34]. In this study, the relative contributions of phytoplankton community were calculated by the CHEMTAX program using the surface pigment concentrations and previously published initial ratios of pigments to Chl. a [32, 33]. The initial pigment/Chl.a ratios used in this study was described in Supplementary Table S1.”

Methods

Point 2: Lines 130 to 131- How do the authors estimate the light attenuation to collect water samples in 50% to 1% light depth to perform in situ incubations?

Response 2: We measured the light intensity at 0.5 m interval water depths from the surface to the bottom. The light extinction coefficients were calculated from the light intensity data at the corresponding water depths. And we added the following sentence in the revised manuscript (Line: 8385). Thanks

Revision 2:

“Light intensity was measured with a photosynthetically active radiation (PAR) quantum sensor (LI-1500, LI-COR, Lincoln, NE, USA).”

“Light intensity was measured at 0.5 m interval water depths from the surface to the bottom with a photosynthetically active radiation (PAR) quantum sensor (LI-1500, LI-COR, Lincoln, NE, USA), and light extinction coefficients (LEC) were calculated from PAR by depth.”

Point 3: Please inform the total PAR and the attenuation for each incubation in the manuscript. I suggest a table with these data.

Response 3: We added supplementary Table S3 (revised) to inform light extinction coefficients, euphotic depth, and secchi depth.

Revision 3: Table S3. Light extinction coefficients, euphotic depth, and secchi depth in the Nakdong River

Point 4: I suggest to rewrite the primary productivity methodology including this information (PAR, light attenuation coefficients).

Response 4: We added the measurement method of light extinction coefficients in 2.1. section (Line: 8385). In addition, we revised primary productivity methodology (Line: 103118). Thanks

Revision 4:

“Light intensity was measured with a photosynthetically active radiation (PAR) quantum sensor (LI-1500, LI-COR, Lincoln, NE, USA).”

“Light intensity was measured at 0.5 m interval water depths from the surface to the bottom with a photosynthetically active radiation (PAR) quantum sensor (LI-1500, LI-COR, Lincoln, NE, USA), and light extinction coefficients (LEC) were calculated from PAR by depth.”

“Primary productivity measurements were conducted through in-situ incubation experiments using a 13C tracer once or twice a month from April to October 2018. Water samples (500 mL) were filtered through a 300 μm mesh to remove mesozooplankton and transferred to polycarbonate incubation bottles (500 mL) cleaned with HCl and deionized water. Bottles were covered with neutral density screens (LEE filters, Andover, UK) to match the corresponding light intensity (100%, 50%, 12%, and 1%) to the surface irradiances [31]. After adding labeled carbon stable isotope tracer (NaH13CO3, 98 atom %, Sigma-Aldrich, ST Louis, MO, USA) to increase the 13C atomic percent of DIC in the incubation bottles to around 10%, bottles were incubated in-situ for 4 h. After incubation, samples (100 mL) were filtered through pre-combusted GF/F filters. Filters were preserved at –80℃ until analysis. After lyophilization and addition of 12 M HCl fumes to remove inorganic carbon for 12 h, POC concentrations and the carbon stable isotope ratio were measured using a Finnigan Delta+XL mass spectrometer at the ASIF. Carbon uptake rates of phytoplankton were calculated using the equation of Hama et al. [32].”

“Primary productivity was measured thorough in-situ incubation experiment using 13C as a tracer. This method directly measures the amount of carbon assimilated into phytoplankton cells. In addition, in-situ incubation is allowed in the natural aquatic environments because 13C tracer has no hazardous radioactive [22]. Water samples (500 mL) collected from each relative light depths (100%, 50%, 12%, and 1%) were filtered through a 300 μm mesh to remove mesozooplankton and transferred to polycarbonate incubation bottles (500 mL) cleaned with HCl and deionized water. The incubation bottles were covered with neutral density screens (LEE filters, Andover, UK) to match the corresponding relative light intensity (100%, 50%, 12%, and 1%) to the surface irradiances [23]. The labeled carbon stable isotope tracer (NaH13CO3, 98 atom %, Sigma-Aldrich, ST Louis, MO, USA) was added to increase the 13C atomic percent of DIC in the incubation bottles to approximately 10%, and the bottles were incubated for 4 h. After the incubation, the water samples (100 mL) were filtered through pre-combusted GF/F filters. Filters were preserved at –80℃ until analysis. After lyophilization, inorganic carbon was removed by 12 M HCl fuming (12 h). POC concentrations and the carbon stable isotope ratio were measured using a Finnigan Delta+XL mass spectrometer at the Alaska Stable Isotope Facility (ASIF, University of Alaska Fairbanks, Fairbanks, USA). Carbon uptake rates of phytoplankton were calculated using the equation of Hama et al. [22].”

Point 5: Lines 164-165- please inform the methodology to qualitative and quantitative phytoplankton analyses under microscopy.

Response 5: We just compared the microscopic data to confirm the application of the CHEMTAX program in Nakdong River environments. The method of qualitative and quantitative phytoplankton analyses under microscopy was described by Kim et al. (2019), and we used the published data set.

Kim, S.; Chung, S.; Park, H.; Cho, Y.; Lee, H. Analysis of environmental factors associated with cyanobacterial dominance after river weir installation. Water (Switzerland) 2019, 11(6), 1163.

Results

Point 6: Lines 2-20- I suggest to include PAR data here, as well as vertical variation in PP. PE curves can be added here, and parameters such as alfa and Pmax calculated.

Response 6: Thanks for the Reviewer #1’s suggestion. Unfortunately, PI curves did not show significant correlation except for ND-1 (Pmax = 843.0) and ND-2 (Pmax = 892.0). In this study, it seems that various environment factors besides light intensity affected primary production besides. We determine that the PI curves were not discussed in manuscript. Instead, the figures of PI curve were shown for

Reviewer #2.

Point 7: Suggestion: Present the PCA results in this section.

Response 7: We have used PCA result based on acquired data to discuss environmental factors controlling primary production. So we would like to show PCA results in discussion section (Figure 6).

Discussion

Point 8: Line 65- Primary production was high in spring and summer (Figure 2a)- This sentence must be in the Results section.

Response 8: We erased this sentence in discussion section, and it was already revised this sentence in results sections (Line: 236). Thanks.

Point 9: First paragraph- lines 65 to 75- It is difficult to compare PP values without PAR values. I suggest to include PAR variation, or euphotic zone depth variation Zeu (Secchi). Higher values during summer and spring can be related to light availability.

Response 9: We already performed PCA including light intensity (PAR), euphotic depth, and secchi depth (Figure 6). Sorry for error with uploading figure. The result of PCA showed that light intensity was positively loaded in summer than spring.

Points 10_ Suggestion: If PAR data or Zeu depth data are available, including at least one of these variables to the PCA can be interesting, specially considering the periods with high cryptophytes densities.

Response 10: We performed PCA visualized with biplot to identify the environmental factors loaded positively or negatively depending on the seasons. PCA showed that the light intensity was negatively loaded on the basis of PC1 and the euphotic depth and sechhi depth were negatively loaded on the basis of PC2 in autumn, when the contribution of cryptophytes was relatively high (Figure 6).

Conclusion

Very clear and well written.

Reviewer 2 Report

First of all, I just want to remark two difficulties in the revision unrelated to the contents. The tables are inserted in the text, but there are no figures in the PDF file (only the figure legends). Where are the figures of the main text? The supplementary file only contains the supplementary tables and figures. In addition, the continuous line numbering of the PDF file is present only in the first pages.

This study describes the relationship between the physico-chemical variables and the phytoplankton in a river, with an especial focus on the factors that triggers the blooms of potentially harmful species. There are two main deficiencies:

  • The period of sampling is short, only 7-8 months and this does not cover the annual seasonality. If you only sampled in October, this does not cover the full autumn season.
  • The percentage of an accessory pigment versus total Chl a is not the phytoplankton abundance. Microscopic observations are necessary.

There are more supplementary figures than figures in the main text. This is an electronic journal, and there are less restrictions in space. Please consider to place some of the supplementary figures in the main text (if it is justified).

The English grammar and style is correct.

Minor comments:

Abstract. line 17. "To" in bold type?

line 17: To identify.. favorable environmental conditions for harmful algal blooms (HABs)

This sounds bad. It looks like that you want to find favorable conditions to have HABs.

Better,  To identify the factors that trigger the harmful algal blooms (HABs)

line 24: spring (1,130 ± 1,140 23 mg C m−2 d−1) and summer (1,060 ± 814 mg C m−2 d−1) than autumn (180 ± 220 mg C m−2 d−1),

Autumn is from late September to late December. October is only the beginning of the autumn. Then, please report the month.

Introduction

The first paragraphs of the introduction are too general. The introduction can begin in the line 59: The Nakdong River is the longest river in South Korea…

line 38: hydraulic retention time (HRT),

This term needs a definition when first cited in the text.

line 48: Phytoplankton community composition has been determined by microscopic analysis as well as high-performance liquid chromatography (HPLC) analysis.

This is the introduction, not the M&M. Where is the description of the microscope observations methods and results?

line 50: picophytoplankton and minor phytoplankton groups can be analyzed

Confusing, picophytoplankton is also a minor phytoplankton groups. Do you mean nanoplankton?

Methods

line 99; inorganic nutrients such as ammonium (NH4+), nitrate + nitrite (NO3- + NO2-),

Please check the height of the symbols + and -.

line 107: In-situ incubation experiment was performed to measure primary productivity using 13C as a tracer.

Here, you introduce the 13C, but the method is explained later in other section.

line 160: pigment to Chl. a

Please check if the abbreviation is Chl a (the dot maybe is not necessary)

line 163: The relative contributions of the phytoplankton community... were compared to the microscopy results.

Again, where are the microscopic observations?

The continuous line numbering is lost.

Page 10. line 26: variations in accessory pigment concentrations were similar at all study sites. Fucoxanthin (a marker pigment for diatoms)...

There are freshwater dinoflagellates (Peridiniopsis, Durinskia) with fucoxanthin. For that reason, the microscopic observations are important.

Page 10, line 45:    the contributions of diatoms to the total phytoplankton classes ranged from 0 to 96% (46 ± 32%), cryptophytes from 2 to 64% (20 ± 16%), 46 chlorophytes from 0 to 76% (17 ± 18%), and cyanobacteria from…

Wrong, these are the percentages of the accessory pigment when compared to the total chlorophyll a. This is not the percentage of abundance of each group of phytoplankton.

Page 11, line 63, Heading: Seasonal and spatial variations in primary production in the regulated water environment of a continuous weir system

Please consider a shorter heading title.

I am unable to find the five figures of the main text. Please consider to place some of the supplementary figures or tables in the main text.

Author Response

We appreciate for the Reviewer 2 comments.

We completed the answers to the comments and suggestions of Reviewer 2, and revised the manuscript accordingly.

In addition, we wrote the “Authorship change form” due to added one author.

Response to Reviewer 2 Comments

Reviewer #2

General points:

First of all, I just want to remark two difficulties in the revision unrelated to the contents. The tables are inserted in the text, but there are no figures in the PDF file (only the figure legends). Where are the figures of the main text? The supplementary file only contains the supplementary tables and figures. In addition, the continuous line numbering of the PDF file is present only in the first pages.

Response: We are really sorry for the Reviewer #2’s confusion. It seems that there were problems with uploading figures and line number due to an error on the server. It is our fault that we have not carefully checked when submitting earlier. We have made all corrections based on the Reviewer #2’s points. Thanks a lot. Number of reference, figure, and table were changed while revising the manuscript. So, Figure number and table number in point may be different from those in response. Please also check the newly attached Figures and Tables.

This study describes the relationship between the physico-chemical variables and the phytoplankton in a river, with an especial focus on the factors that triggers the blooms of potentially harmful species. There are two main deficiencies:

  1. The period of sampling is short, only 7-8 months and this does not cover the annual seasonality. If you only sampled in October, this does not cover the full autumn season.

Response: We could not cover whole autumn season due to the limitation of time available for the in-situ incubation experiment. Furthermore, our main research purpose focused on the HABs season from the spring to summer. So we had discussed spatial and temporal variability of the primary production instead of annual seasonality in the text.

  1. The percentage of an accessory pigment versus total Chl a is not the phytoplankton abundance. Microscopic observations are necessary.

Response: Phytoplankton abundance was not estimated by the percentage of an accessory pigment versus total Chl. a. It was calculated by the CHEMTAX program based on the pigment/Chl. a ratios (Line: 155164). We would like to apply the CHEMTAX program to freshwater environments. Phytoplankton community composition calculated by the CHEMTAX program showed a good correlation with cell numbers measured by microscopy, except for cryptophytes (Supplementary Figure S4). This finding supported that the CHEMTAX program based on pigment analysis is feasible to freshwater environment and able to describe to phytoplankton community composition well.

There are more supplementary figures than figures in the main text. This is an electronic journal, and there are less restrictions in space. Please consider to place some of the supplementary figures in the main text (if it is justified).

Response: We transferred Figures showing the concentrations of nutrients (existing Figures S3, S4, S5, and S6) to main text (revised Figure 2). Instead, we erased existing Table 2 representing the concentrations of nutrients. Thanks.

The English grammar and style is correct.

Specific points:

Abstract

Point 1: line 17. "To" in bold type?

Response 1: It seems that there was a problem in the process of converting to PDF. We revised bold type letter (Line: 20).

Point 2: line 17: To identify. favorable environmental conditions for harmful algal blooms (HABs). This sounds bad. It looks like that you want to find favorable conditions to have HABs. Better, To identify the factors that trigger the harmful algal blooms (HABs).

Response 2: Thanks for your helpful points. We revised this sentence as your suggestion (Line: 20).

Revision 2:

“To identify key factors controlling primary production and favorable environmental conditions for harmful algal blooms (HABs),”

“To identify key factors that control primary production and trigger cyanobacterial harmful algal blooms (cHABs),”

Point 3: line 24: spring (1,130 ± 1,140 23 mg C m−2 d−1) and summer (1,060 ± 814 mg C m−2 d−1) than autumn (180 ± 220 mg C m−2 d−1), Autumn is from late September to late December. October is only the beginning of the autumn. Then, please report the month.

Response 3: Abstract have limitation of the number of words, so we could not specify the month according to season. But it was mentioned in main text. Thanks.

Introduction

Point 4: The first paragraphs of the introduction are too general. The introduction can begin in the line 59: The Nakdong River is the longest river in South Korea… line 38: hydraulic retention time (HRT), This term needs a definition when first cited in the text, line 48: Phytoplankton community composition has been determined by microscopic analysis as well as high-performance liquid chromatography (HPLC) analysis. This is the introduction, not the M&M. Where is the description of the microscope observations methods and results?

Response 4: We erased the general introduction and revised it entirely (Line: 3972). And we transferred this paragraph about the CHEMTAX program from introduction to methodology section (Line: 154166).

Revision 4:

Introduction

“Primary productivity is controlled by various physical, chemical, and biological factors [1]. Previous studies demonstrated that primary productivity can be affected by light intensity, water temperature, hydraulic retention time (HRT), and nutrients [2–8]. Phytoplankton community composition can be also influenced by these factors [3, 9, 10]. Phytoplankton could contribute extremely to primary productivity in aquatic environments where the distributions of attached algae and aquatic plants are limited due to water-level fluctuations caused by weirs or dams [1]. Anthropogenic factors including climate change and water use significantly change the amplitude of water-level fluctuations, causing strong disturbances in environmental conditions of lakes [11]. Water-level fluctuations influence phytoplankton community composition and densities by affecting the amount of suspended sediments, light availability, and diluting biomass [6, 10].

Algal blooms deteriorate water quality by increasing organic matter and microbial activity [4]. Microbial decomposition activity causes serious problems such as increased oxygen consumption and dissolved oxygen depletion [12]. In particular, the increase in cyanobacterial harmful algal blooms (cHABs) caused by climate change and nutrient inputs in recent decades is of global concern [4]. Proliferation of cyanobacteria changes the color of the water, creates scums and odors, and produces harmful toxins (e.g. microcystins, MCs) that are hazardous to drink and can potentially affect aquatic organisms [13, 14]. Therefore, primary productivity and phytoplankton community composition need to be investigated in the regulated water environment including continuous dams or weirs.

The Nakdong River is the longest river in South Korea with a length of 525 km and watershed area of 23,384 km2. It is one of the largest water resources for drinking, industry, and agriculture [15]. Areas surrounding the middle and downstream are densely populated, and industrial complexes are concentrated near by the Nakdong River [16]. The Korean government conducted the ‘Four Major Rivers Project’ from 2009 to 2012 to solve repeated flooding and droughts [17]. Eight continuous weirs were constructed over approximately 200 km on the main stream. After construction of the weirs, there was an increase in HRT, water level, and storage volume [18]. In addition, total phosphorus (TP) concentrations decreased noticeably from 2001 to 2016, particularly after 2013 due to reinforcement of water quality regulations [19]. Since the 1990s, diatom blooms have been reported in spring and winter, and cyanobacterial blooms in summer [20]. After the weir construction, the cell density of diatoms decreased, while cyanobacteria abundance increased [15].

Few previous studies have reported spatial and temporal variations in primary production, including cHABs, in a continuous weir system like the one in the Nakdong River. Our objectives in this study were to understand the key environmental factors controlling spatio-temporal variability in primary production, and to identify environmental conditions causing cHABs in a continuous weir system in the Nakdong River.”

2.4. Phytoplankton pigment analysis using HPLC and application of the CHEMTAX program

“Pigment analysis using HPLC has the advantages of providing relatively reliable and consistent results in a short time. In addition, pico-sized phytoplankton and minor phytoplankton groups can be analyzed at the class level [28, 29]. Cyanobacteria have very small cell sizes and aggregate to form colonies, making it time-consuming to enumerate cyanobacterial cells in the freshwater environment [30]. However, quantitative and qualitative measurements of phytoplankton marker pigments have limitations in estimating the phytoplankton community and distribution. Mackey et al. [31] developed the CHEMTAX program to assess the contribution of each community to phytoplankton biomass using a steepest descent algorithm and factor analysis based on pigment/Chl. a ratios. The CHEMTAX program was developed for marine phytoplankton research, but has recently been applied to various freshwater environments [30, 32–34]. In this study, the relative contributions of phytoplankton community were calculated by the CHEMTAX program using the surface pigment concentrations and previously published initial ratios of pigments to Chl. a [32, 33]. The initial pigment/Chl.a ratios used in this study was described in Supplementary Table S1.”

Point 5: line 50: picophytoplankton and minor phytoplankton groups can be analyzed~. Confusing, picophytoplankton is also a minor phytoplankton groups. Do you mean nanoplankton?

Response 5: Minor group means the phytoplankton that are not dominant such as cryptophytes.

Methods

Point 6: line 99; inorganic nutrients such as ammonium (NH4+), nitrate + nitrite (NO3- + NO2-), Please check the height of the symbols + and -.

Response 6: It was confirmed that the heights of the symbols + and – were correct.

Point 7: line 107: In-situ incubation experiment was performed to measure primary productivity using 13C as a tracer. Here, you introduce the 13C, but the method is explained later in other section.

Response 7: As the Reviewer #2’s advice, we corrected the order of the paragraphs to describe in-situ experiment using 13C tracer after the sample preparation paragraph. In addition, we revised primary productivity methodology (Line: 102118).

Revision 7:

2.3. In-situ incubation experiment for primary productivity

“Primary productivity measurements were conducted through in-situ incubation experiments using a 13C tracer once or twice a month from April to October 2018. Water samples (500 mL) were filtered through a 300 μm mesh to remove mesozooplankton and transferred to polycarbonate incubation bottles (500 mL) cleaned with HCl and deionized water. Bottles were covered with neutral density screens (LEE filters, Andover, UK) to match the corresponding light intensity (100%, 50%, 12%, and 1%) to the surface irradiances [31]. After adding labeled carbon stable isotope tracer (NaH13CO3, 98 atom %, Sigma-Aldrich, ST Louis, MO, USA) to increase the 13C atomic percent of DIC in the incubation bottles to around 10%, bottles were incubated in-situ for 4 h. After incubation, samples (100 mL) were filtered through pre-combusted GF/F filters. Filters were preserved at –80℃ until analysis. After lyophilization and addition of 12 M HCl fumes to remove inorganic carbon for 12 h, POC concentrations and the carbon stable isotope ratio were measured using a Finnigan Delta+XL mass spectrometer at the ASIF. Carbon uptake rates of phytoplankton were calculated using the equation of Hama et al. [32].”

2.2. In-situ incubation experiment for primary productivity

“Primary productivity was measured thorough in-situ incubation experiment using 13C as a tracer. This method directly measures the amount of carbon assimilated into phytoplankton cells. In addition, in-situ incubation is allowed in the natural aquatic environments because 13C tracer has no hazardous radioactive [22]. Water samples (500 mL) collected from each relative light depths (100%, 50%, 12%, and 1%) were filtered through a 300 μm mesh to remove mesozooplankton and transferred to polycarbonate incubation bottles (500 mL) cleaned with HCl and deionized water. The incubation bottles were covered with neutral density screens (LEE filters, Andover, UK) to match the corresponding relative light intensity (100%, 50%, 12%, and 1%) to the surface irradiances [23]. The labeled carbon stable isotope tracer (NaH13CO3, 98 atom %, Sigma-Aldrich, ST Louis, MO, USA) was added to increase the 13C atomic percent of DIC in the incubation bottles to approximately 10%, and the bottles were incubated for 4 h. After the incubation, the water samples (100 mL) were filtered through pre-combusted GF/F filters. Filters were preserved at –80℃ until analysis. After lyophilization, inorganic carbon was removed by 12 M HCl fuming (12 h). POC concentrations and the carbon stable isotope ratio were measured using a Finnigan Delta+XL mass spectrometer at the Alaska Stable Isotope Facility (ASIF, University of Alaska Fairbanks, Fairbanks, USA). Carbon uptake rates of phytoplankton were calculated using the equation of Hama et al. [22].”

Point 8: line 160: pigment to Chl. a Please check if the abbreviation is Chl a (the dot maybe is not necessary)

Response 8: Thanks for the Reviewer #2’s point. The full name of Chl. a was written when it was mentioned at first. As the sentence order had been changed, the sentence that were mentioned Chl. a at first was changed. However, considering your suggestion, Chl. a is an abbreviation for Chlorophyll a, so we thought dot is necessary (Line: 146148). Thanks.

Revision 8:

“The absorbance of each pigment was detected at 440 nm. Each peak was distinguished according to retention time and a spectrum of standards for Chl. a (Sigma-Aldrich, ST Louis, MO, USA) and other pigments (DHI water & Environment, Hørsholm, Denmark).”

“The absorbance of each pigment was detected at 440 nm. Each peak was distinguished according to retention time and a spectrum of standards for Chlorophyll a (Chl. a; Sigma-Aldrich, ST Louis, MO, USA) and other pigments (DHI water & Environment, Hørsholm, Denmark).”

Point 9: line 163: The relative contributions of the phytoplankton community... were compared to the microscopy results. Again, where are the microscopic observations?

Response 9: We just compared the microscopic data to confirm the application of the CHEMTAX program in Nakdong River environments. The method of qualitative and quantitative phytoplankton analyses under microscopy was described by Kim et al. (2019), and we used the published data set. The results of microscopic were used to confirm the validity of the phytoplankton community composition calculated by the CHEMTAX program. The relationship between the results calculated by the CHEMTAX program and analyzed by microscopy was shown in Figure S4.

Kim, S.; Chung, S.; Park, H.; Cho, Y.; Lee, H. Analysis of environmental factors associated with cyanobacterial dominance after river weir installation. Water (Switzerland) 2019, 11(6), 1163.

Results

Point 9: Page 10. line 26: variations in accessory pigment concentrations were similar at all study sites. Fucoxanthin (a marker pigment for diatoms). There are freshwater dinoflagellates (Peridiniopsis, Durinskia) with fucoxanthin. For that reason, the microscopic observations are important.

Response: We absolutely agree with your points on the importance of microscopic observations because phytoplankton species composition can be clearly identified, providing better information to understand the dynamics of specific phytoplankton species. However, in this study, we tried to confirm the variations in phytoplankton community composition through the CHEMTAX program based on empirical pigment composition ratio in freshwater environments (Paerl et al., 2015). As you mentioned, even though phytoplankton have common pigments, the advantage of the CHEMTAX program is that it can determine the contribution of each phytoplankton class to total phytoplankton abundance using initial pigment/Chl. a ratio considering common pigments of phytoplankton. The information of the CHEMTAX program is detailed in methodology section (Line: 154166). In addition, we did not discuss the factors affecting variations in phytoplankton species, and we just refer to it when explaining that cyanobacteria dominated in summer were mostly toxin-producing species. The results of phytoplankton species composition were mainly published to Kim et al (2019). Thanks a lot.

Paerl, H.W.; Xu, H.; Hall, N.S.; Rossignol, K.L.; Joyner, A.R.; Zhu, G.; Qin, B. Nutrient limitation dynamics examined on a multi-annual scale in Lake Taihu, China: Implications for controlling eutrophication and harmful algal blooms. J. Freshw. Ecol. 2015, 30(1), 5–24.

Kim, S.; Chung, S.; Park, H.; Cho, Y.; Lee, H. Analysis of environmental factors associated with cyanobacterial dominance after river weir installation. Water (Switzerland) 2019, 11(6), 1163.

Point 10: Page 10, line 45: the contributions of diatoms to the total phytoplankton classes ranged from 0 to 96% (46 ± 32%), cryptophytes from 2 to 64% (20 ± 16%), 46 chlorophytes from 0 to 76% (17 ± 18%), and cyanobacteria from… Wrong, these are the percentages of the accessory pigment when compared to the total chlorophyll a. This is not the percentage of abundance of each group of phytoplankton.

Response 10: Phytoplankton community composition was calculated by the CHEMTAX program based on the combination of empirical pigment/Chl. a ratios. It was not estimated only the percentage of pigment/Chl. a ratios. The information of the CHEMTAX program is detailed in method section (Line: 154166). Thanks.

Discussion

Point 11: Page 11, line 63, Heading: Seasonal and spatial variations in primary production in the regulated water environment of a continuous weir system. Please consider a shorter heading title.

Response 11: Very thank for the Review #2’s suggestion. We revised subtitle shorter as follows (Line: 282). Thanks.

Revision 11:

“4.1. Seasonal and spatial variations in primary production in the regulated water environment of a continuous weir system”

“4.1. Seasonal and spatial variations in primary production in a continuous weir system”

Point 12: I am unable to find the five figures of the main text. Please consider to place some of the supplementary figures or tables in the main text.

Response 12: We uploaded main figures after checking carefully in this time. Sincerely sorry for failure uploaded figures. We transferred Figures showing the concentrations of nutrients (existing Figure S3, S4, S5, and S6) to main text (revised Figure 2). Instead, we erased existing Table 2 representing the concentrations of nutrients. Very thanks for the Review #2’s suggestion.

Reviewer 3 Report

The research may be of interest, but It's far from achieving its objectives, they use a large number of physical, chemical and biological variables that are not used or interpreted in the end, similarly, it occurs with the large number of statistical analyzes used, that don't show relevance in the results presented. Some tables and figures mentioned in the manuscript are not available.

Author Response

We appreciate for the Reviewer 3 comments.

We completed the answers to the comments and suggestions of Reviewer 3, and revised the manuscript accordingly.

In addition, we wrote the “Authorship change form” due to added one author.

Response to Reviewer 3 Comments

Reviewer #3

General points:

The research may be of interest, but It's far from achieving its objectives, they use a large number of physical, chemical and biological variables that are not used or interpreted in the end, similarly, it occurs with the large number of statistical analyzes used, that don't show relevance in the results presented. Some tables and figures mentioned in the manuscript are not available.

The introduction is poor and doesn't mention all the background on Primary Production or HABs, the methodology is confusing and messy, in addition a large number of bibliographic references are needed, the results need more synthesis and analysis, and the discussion needs to deepen in the relationship of the physicochemical variables with the meteorological and hydrological variables.

Many changes are required that I propose below.

Response: The authors greatly thanks the Reviewer #3 for reviewing our manuscript. We are really sorry for the Reviewer #3’s confusion. It seems that there were problems with uploading figures and line number due to an error on the server. It is our fault that we have not carefully checked when submitting earlier. The manuscript including figures and tables have been carefully revised to address and reflect the Reviewer #3’s points and suggestions. Number of reference, figure, and table were changed while revising the manuscript. So, Figure number and table number in point may be different from those in response. Please also check the newly attached Figures and Tables. Thanks a lot.

Specific points

Abstract

Point 1: The abstract is a short summary of the manuscript, it must have the same parts as the manuscript, it must have an introduction, methodology, results and conclusions, and all these elements must be very well linked. I would like to see a restructuring of the abstract.

Response 1: We revised abstract to include what should have into the abstract such as introduction, methodology, results, and conclusions (Line: 2032).

Revision 1:

Introduction: To identify key factors that control primary production (P.P.) and trigger cyanobacterial harmful algal blooms (cHABs)

Methodology: we investigated spatio-temporal variations in P.P. and phytoplankton community composition in a continuous weir system in the Nakdong River once or twice a month from April to October 2018. P.P. was measured through in-situ incubation experiment using a 13C tracer.

Results: Relative contributions of the phytoplankton community were calculated by the CHEMTAX program based on pigment analysis using a high-performance liquid chromatography (HPLC). P.P. was higher in spring (1,130 ± 1,140 mg C m-2 d-1) and summer (1,060 ± 814 mg C m-2 d-1) than autumn (180 ± 220 mg C m-2 d-1), and tended to increase downstream. P.P. was negatively related to PO43- (r = -0.41, p < 0.01) due to utilization by phytoplankton during the spring and summer when it was high. The relative contribution of cyanobacteria (mainly Microcystis sp. and Aphanizomenon sp.) was positively correlated with water temperature (r = 0.79, p < 0.01) and hydraulic retention time (HRT, r = 0.67, p < 0.01)

Conclusions: suggesting that these two factors should affect cHABs in summer. Therefore, to control HRT could be one of solutions for reducing cHABs in a continuous weir system.”

Introduction

Point 2: The paragraph on lines 59 to 74 could be in a section called "study area" at the beginning of the methodology. It's out of place in the current position

Response 2: This paragraph provides a general description of the Nakdong River. In methodology section, we explained specific sampling sites.

Point 3: The introduction needs to talk about 13C and its use to measure Primary Productivity

Response 3: We added an information about the measurement of primary productivity using 13C tracer in methodology section (Line: 103106) to help understanding this method.

Revision 3:

“Primary productivity was measured thorough in-situ incubation experiment using 13C as a tracer. This method directly measures the amount of carbon assimilated into phytoplankton cells. In addition, in-situ incubation is allowed in the natural aquatic environments because 13C tracer has no hazardous radioactive [22].”

Point 4: Line 92. A map with the location of the weir is needed

Response 4: The figures were not uploaded when we submit. We checked carefully. Sincerely sorry about that. We checked carefully and uploaded all figures and tables in this time.

Point 5: Line 93 – 111. Here the analyzes that were carried out must be very clear. Everything is very messy; they should separate by type of analysis and differentiate the processes carried out very well. In addition, bibliographic references are required.

Response 5: We revised methodology 2.1 section entirely to explain how to carry out sampling more accurate (Line: 83100). Thanks.

Revision 5:

“Light intensity was measured at 0.5 m interval water depths from the surface to the bottom with a photosynthetically active radiation (PAR) quantum sensor (LI-1500, LI-COR, Lincoln, NE, USA), and light extinction coefficients (LEC) were calculated from PAR by depth. The 1% depth of relative light intensity was determined as the euphotic depth. Water temperature (℃) and pH were measured using a Hydrolab DS5X Multiparameter sonde (OTT Hydromet, Loveland, CO, USA). Water samples (approximately 4 L) from four water depths (100%, 50%, 12%, and 1% surface irradiances) were collected using a Van Dorn sampler. To analyze dissolved inorganic nutrients, dissolved inorganic carbon (DIC), particulate organic carbon (POC), and particulate nitrogen (PN), surface water samples were filtered using pre-combusted (450℃, 4 h) GF/F filters (Whatman, Maidstone, UK). Filtrates for analysis of ammonium (NH4+), nitrate + nitrite (NO3- + NO2-), phosphate (PO43-), and silicate (SiO2) were put into 125 mL high density polyethylene (HDPE) bottles (Nalgene, Rochester, New York) cleaned with HCl and deionized water (Aquapuri 5 series, Young In Scientific, Korea). Mercury chloride (HgCl2; Sigma-Aldrich, ST Louis, MO, USA) was added to suppress microbial activity [21], and then transferred to the laboratory in a frozen state. Filtrates for DIC samples were also put into 40 mL amber vial with HgCl2. Filter samples for analysis of POC and PN were stored frozen at –80℃ until analysis. Water samples (500 mL) for pigment analysis were filtered through pre-combusted GF/F filters. Filters were wrapped in aluminum foil to prevent photolysis and then transferred to the laboratory in a frozen state. They were stored at –80℃ until further analysis.”

Point 6: Line 114 – 123. It is confusing, it must be separated and presented in parts, one or more bibliographic references on Dissolved Inorganic Carbon must be cited

Response 6: This section was written according to the order of sampling. We revised this paragraph entirely to help understanding. And DIC was analyzed by TOC analyzer based on standard calibration curve. Thanks for the Reviewer #3’s point.

Point 7: Line 125 – 138. Based on which method did you carry out the incubation and sampling?

Response 7: We carried out in-situ incubation experiment using 13C tracer suggested by Hama et al. (1983). We added this information to methodology section (Line: 103106).

Revision 7:

“Primary productivity was measured thorough in-situ incubation experiment using 13C as a tracer. This method directly measures the amount of carbon assimilated into phytoplankton cells. In addition, in-situ incubation is allowed in the natural aquatic environments because 13C tracer has no hazardous radioactive [22].”

Point 8: Line 174. I don't see the correlation in table S3, only sampling dates.

Response 8: Samples for pigment analysis and microscopic analysis were collected in different dates. Therefore, the information about sampling dates were indicated in Table S3, and we referred that in Table caption as well as manuscript. The correlation between the results of the CHEMTAX program and microscopy was showed in revised Figure S4 (Line: 173177).

Point 9: Line 140-165. What are the methodologies used for HPLC and CHEMTAX?

Response 9: We analyzed phytoplankton pigment using HPLC with UV-Vis detector, subsequently the CHEMTAX program was used to calculate the contribution of phytoplankton community based on pigment results. The detailed methods were mentioned in 2.4 (Line: 154166).

Results

Point 10: Line numbering is lost from line 198

Response 10: We are really sorry for your confusion. It seems that there were problems with uploading figures and line number due to an error on the server. It is our fault that we could not check carefully when submitting. We have made all corrections.

Point 11: Page 7. I would like to see graphs S3, S4, S5 and S6 within the manuscript.

Response 11: As the Reviewer #3’s point, we transferred Figures representing the concentrations of nutrients (existing Figure S3, S4, S5, and S6) to main text (revised Figure 2). Instead, we erased existing Table 2. Thanks for the Reviewer #3’s suggestion.

Point 12: Page 9. There is talk of the Euphotic Zone and 1% of the available light in depth, but they have not said how to obtain this Euphotic Zone. The authors did say that the measurements were made with an LI-COR device, but how did they convert the light values from the device to Optical Depths?

Response 12: We measured the light intensity at 0.5 m intervals from the surface to the bottom. The light extinction coefficients were calculated from the light intensity data by depth, and the depth showing 1% relative light intensity compared to the surface was determined as euphotic depth. And we added this information in manuscript (Line: 8385).

Revision 12:

“Light intensity was measured with a photosynthetically active radiation (PAR) quantum sensor (LI-1500, LI-COR, Lincoln, NE, USA).”

“Light intensity was measured at 0.5 m interval water depths from the surface to the bottom with a photosynthetically active radiation (PAR) quantum sensor (LI-1500, LI-COR, Lincoln, NE, USA), and light extinction coefficients (LEC) were calculated from PAR by depth.”

Point 13: How deep is 1% of available light in meters?

Response 13: The euphotic depth (1% relative light depth) ranged from 1.3 m to 8.0 m (mean ± stdev = 3.5 ± 1.4 m). We added the supplementary table (Table S3) described light extinction coefficients, euphotic depth, and secchi depth.

Discussion

Point 14: Page 12 Line 107. Figure 5 I suppose is table S5, in the manuscript it says PCA and in the table it says Spearman's Correlation, what is correct?

Response 14: We conducted PCA, but figure was not uploaded. PCA showed relationships among environmental factors. We uploaded that figure (Figure 6). Spearman’s correlation analysis (Figure S5) was additionally performed to confirm their significant correlations.

Point 15: Also in this table S5 I see the variable “Secchi Disk”, they have not talked about that.

Response 15: We added the supplementary table described light extinction coefficients, euphotic depth, and secchi depth (Table S3).

Point 16: Page 12, Line 127-131. This is a very important part, it should be deepened

Response 16: We added a sentence to discuss more and added our result in this paragraph. Thanks the Reviewer #3’s suggestion.

Revision 16:

“The inflow of PO43- can promote primary productivity in aquatic environments where phosphate is a limiting factor [33, 43, 44]. Furthermore, excess PO43- concentrations and increasing N/P ratio can induce severe cyanobacterial blooms [4].”

“In addition, LEC was the highest, and euphotic depth and secchi depth was the lowest due to turbidity at that time (Supplementary Table S3).”

Point 17: Page 13, Line 142-149. During your study, were there any Harmful Algal Blooms? What were the dates of these HABs?

Response 17: We added the criteria of cHABs in Korea and the sentence including cited results to represent that there were cHABs in summer. Thanks for the Reviewer #3’s point (Line: 361366).

Revision 17:

“The Korean alert system of algal blooms, as criteria for cyanobacteria, have three levels which are Caution (500 cells mL-1), Warning (5,000 cells mL-1), and Outbreak (1,000,000 cells mL-1) [48]. The exceeding 5,000 cells mL-1 of cyanobacteria occurred 69% of all the study sites in summer (from June to August), and the highest cyanobacteria cell density was 1,264,052 cells mL-1 at Hapcheon-Changnyeong weir in August 2018 (Water Environment Information System; http://water.nier.go.kr).”

Ahn, C.-Y.; Joung, S.-H.; Yoon, S.-K.; Oh, H.-M. Alternative alert system for cyanobacterial bloom, using phycocyanin as a level determinant. J. Microbiol. 2007, 45(2), 98–104.

Point 18: What is the criteria used to differentiate a HAB from an Algal Bloom? What is the criterion can be said that there is a phytoplankton bloom, that is, what is the difference between the normal behavior of phytoplankton and an behavior outside of normal?

Response 18: Phytoplankton are important primary producer providing organic matters. However, algal blooms trigger the depletion of oxygen in water system. Among various phytoplankton community, cyanobacterial cell density continues to increase due the increase in hydraulic retention time as well as water temperature induced in global warming. HABs are a phenomenon in which phytoplankton producing harmful toxins such as cyanobacteria cause blooms. The problems of cyanobacterial harmful algal blooms were mentioned in introduction (Line: 4956).

Point 19: Page 15, Line 26-20. So if the temperature causes blooms, why in Antarctica are there blooms? Will the availability of light have an answer?

Response 19: The availability of light can be an answer. But, our results showed the increase in water temperature affect positively the contribution of cyanobacteria to total phytoplankton in summer, not all phytoplankton communities.

Reviewer 4 Report

Numbering: to make our task less difficult, it is preferable to use continuous numbering instead of starting it on each page or part of the text.

This is a nice and complete study that deserves publication, nevertheless, it needs improvement.

First, in the title, the authors mention HABs when they are really referring to cHABs (cyanobacterial harmful algal blooms). This is very confusing when reading. I suggest changing every HABs mention to cHABs.

Keywords: they lack spaces, making them difficult to read.

Authors should state first how the phytoplankton community is formed, because they treat as the same prokaryotes and eukaryotes, leading to confusion or the idea that they are not well informed.

Figures (not supplementary) were not provided.

Throughout the text: Chl a does not need a period, and a should be in italics, even in the tables (and supplementary tables).

Page (P)2, Line (L) 45-47: This sentence must be re-written.

L48: Phytoplankton community composition has also been studied with molecular tools, such as metagenomics, and not only by microscopy and HPLC analysis. And do not forget that HPLC is a SEPARATION technique, the analysis is performed by the detector (UV-vis, FLD, etc.) of the separated metabolites by chromatography.

L59-74: the authors repeat Nakdong River 5 times in only one paragraph; this should be reviewed, and repetitions prevented.

L70: cyanobacterial algal blooms are not HABs but cHABs.

L80: phytoplankton primary production is obvious. Delete the word phytoplankton. Also, “primary production” is repeated twice in only 2 lines. Writing should be improved.

L102: the authors state to use HgCl2 to suppress microbial activity, but they do not specify where this chemical was bought and a reference to support its use.

L105 and 109: the authors used GF/F filters; their diameter is not as important as the pore size.

L141: lyophilizing time is irrelevant; what matters is if the samples were dry.

L150-151: should be v/v/v.

L157: authors should state what other pigments they are referring to.

Table 1: erase periods from table titles. Specify what you mean by “EL.m”.

On this table and in the rest of the text, it is irrelevant to state exactly the day of maximum and minimum parameters, since it is -of course- the only day in the month it was sampled. This would be important if they sampled every day.

The authors do not explain the lack of information about 2 weirs in Table 1.

All the text: space is missing between the number and °C.

P7L16: the authors mention a “relatively low value”; they need to state these values were relatively low compared to what.

Table 2: same as Table 1: the day of the month is irrelevant, and this information is not useful but makes the table difficult to read.

P9L7-12: These data should be in a table for better visualization. And again, the day only reflects the day of sampling.

P9L16: Clarify what you refer to with “They were…”.

P10L22: this sentence should be re-written.

P10L37-60: there are several grammar mistakes. The whole paragraph should be improved with special attention to the word “contribution” (using singular or plural as needed and trying to avoid so much repetition).

P11L80: the authors refer to the proliferation of algal blooms, but they really mean cyanobacterial blooms.

P12L99: was A major, not an.

P12L122-123: Nakdong River is unnecessarily repeated.

P13L138: the relative contribution was.

L142: Microcystis and Aphanizomenon were the dominant cyanobacterial genera. It is important to state differences, since “algae” or “microalgae” is only a functional and not a taxonomic group.

L145: delete the word cyanobacteria.

L148: upper trophic levels, not positions.

P15l29: harmful cyanobacteria.

P16L30-42: The conclusions are only a repetition of results. This whole part should be improved: should this study be repeated annually to be aware of risks for the ecosystem and human population? Why is this important? What are the authors' recommendations? Why?

References: scientific names are not in italics.

Figures: there were not provided!

Suppl. Mat.: Fig. S1: legend states cubic millimeters when it should be cubic meters.

Fig. S9: probably should not be supplementary but in the main text.

Author Response

We appreciate for the Reviewer 4 comments.

We completed the answers to the comments and suggestions of Reviewer 4, and revised the manuscript accordingly.

In addition, we wrote the “Authorship change form” due to added one author.

Response to Reviewer 4 Comments

Reviewer #4

General points:

Point 1: Numbering: to make our task less difficult, it is preferable to use continuous numbering instead of starting it on each page or part of the text. This is a nice and complete study that deserves publication, nevertheless, it needs improvement.

Response 1: We are really sorry for your confusion. It seems that there were problems with uploading figures and line number due to an error on the server. It is our fault that we have not carefully checked when submitting. We have made all corrections.

Point 2: First, in the title, the authors mention HABs when they are really referring to cHABs (cyanobacterial harmful algal blooms). This is very confusing when reading. I suggest changing every HABs mention to cHABs.

Response 2: We agree with your view point. We revised all HABs to cHABs. Thanks a lot.

Revision 2:

“Key factors controlling primary production and harmful algal blooms (HABs) in a continuous weir system in the Nakdong River, Korea”

“Key factors controlling primary production and cyanobacterial harmful algal blooms (cHABs) in a continuous weir system in the Nakdong River, Korea”

Point 3: Keywords: they lack spaces, making them difficult to read.

Response 3: The process converting to PDF seemed to have problem. We corrected spacing. Sincerely sorry for that.

Point 4: Authors should state first how the phytoplankton community is formed, because they treat as the same prokaryotes and eukaryotes, leading to confusion or the idea that they are not well informed.

Response 4: We did not consider quantitatively prokaryotes and eukaryotes abundances.

Point 5: Throughout the text: Chl a does not need a period, and a should be in italics, even in the tables (and supplementary tables).

Response 5: We thought dot is necessary, because Chl. a is an abbreviation for Chlorophyll a. And a was revised to be in italics. Thanks for checking carefully.

Specific points

Introduction

Point 6: Page (P)2, Line (L) 45-47: This sentence must be re-written.

Response 6: Thanks for the Reviewer #4’s point. We revised this sentence as follows (Line: 5556).

Revision 6:

“Therefore, to investigate phytoplankton primary productivity and their community composition are important in aquatic environments in which dams or weirs have been constructed.”

“Therefore, primary production and phytoplankton community composition need to be investigated in the regulated water environment including continuous dams or weirs.”

Point 7: L48: Phytoplankton community composition has also been studied with molecular tools, such as metagenomics, and not only by microscopy and HPLC analysis. And do not forget that HPLC is a SEPARATION technique, the analysis is performed by the detector (UV-vis, FLD, etc.) of the separated metabolites by chromatography.

Response 7: We agree with the Reviewer #4’s view point. We transferred the information about the CHEMTAX program to methodology section. In addition, the information about detector was added in methodology section (Line: 146148).

Revision 7:

“The absorbance of each pigment was detected at 440 nm.”

“The absorbance of each pigment was detected with UV-visible detector at 440 nm.”

Point 8: L59-74: the authors repeat Nakdong River 5 times in only one paragraph; this should be reviewed, and repetitions prevented.

Response 8: Thanks for the Reviewer #4’s point. We corrected repeated Nakdong River (Line: 5767).

Revision 8:

“The Nakdong River is the longest river in South Korea with a length of 525 km and watershed area of 23,384 km2. It is one of the largest water resources for drinking, industry, and agriculture [21]. Areas surrounding the middle and downstream sections of the Nakdong River are densely populated, and industrial complexes are concentrated nearby the Nakdong River [22]. The Korean government conducted the ‘Four Major Rivers Project’ from 2009 to 2012 to solve repeated flooding and droughts [23]. Eight continuous weirs were constructed over approximately 200 km on the main stream of the Nakdong River. After construction of the weirs, there was an increase in HRT, water level, and storage volume [24]. In addition, total phosphorus (TP) concentrations decreased noticeably from 2001 to 2016, particularly after 2013 due to reinforcement of water quality regulations [25]. Since the 1990s, diatom blooms have been reported in the Nakdong River in spring and winter, and cyanobacterial blooms in summer [26]. After construction of the weirs, the cell density of diatoms decreased, while cyanobacteria abundance increased [21]. The increase in cyanobacterial harmful algal blooms (HABs) caused by climate change and nutrient inputs in recent decades is of global concern [6]. Proliferation of cyanobacteria changes the color of the water, creates scums and odors, and produces harmful toxins (e.g. microcystins, MCs) that are hazardous to drink and can potentially affect aquatic organisms [27, 28].”

“The Nakdong River is the longest river in South Korea with a length of 525 km and watershed area of 23,384 km2. It is one of the largest water resources for drinking, industry, and agriculture [15]. Areas surrounding the middle and downstream are densely populated, and industrial complexes are concentrated near by the Nakdong River [16]. The Korean government conducted the ‘Four Major Rivers Project’ from 2009 to 2012 to solve repeated flooding and droughts [17]. Eight continuous weirs were constructed over approximately 200 km on the main stream. After construction of the weirs, there was an increase in HRT, water level, and storage volume [18]. In addition, total phosphorus (TP) concentrations decreased noticeably from 2001 to 2016, particularly after 2013 due to reinforcement of water quality regulations [19]. Since the 1990s, diatom blooms have been reported in spring and winter, and cyanobacterial blooms in summer [20]. After the weir construction, the cell density of diatoms decreased, while cyanobacteria abundance increased [15].”

Point 9: L80: phytoplankton primary production is obvious. Delete the word phytoplankton. Also, “primary production” is repeated twice in only 2 lines. Writing should be improved.

Response 9: We erased the word phytoplankton and repeated “primary production” (Line: 6872). Thanks.

Revision 9:

“Our objectives in this study were to understand the spatio-temporal variability in phytoplankton primary production, to identify the key environmental factors controlling primary production, and to determine environmental conditions favorable to HABs in a continuous weir system in the Nakdong River.”

“Our objectives in this study were to understand the key environmental factors controlling spatio-temporal variability in primary production, and to identify environmental conditions causing cHABs in a continuous weir system in the Nakdong River.

Methods

Point 10: L102: the authors state to use HgCl2 to suppress microbial activity, but they do not specify where this chemical was bought and a reference to support its use.

Response 10: We added the reference explain that HgCl2 suppress microbial activity. In addition, the information about used HgCl2 was added (Line: 9495).

Revision 10:

“Mercury chloride (HgCl2; Sigma-Aldrich, ST Louis, MO, USA) was added to suppress microbial activity [reference], and then transferred to the laboratory in a frozen state.”

Taipale, S. J.; Sonninen, E. The influence of preservation method and time on the δ13C value of dissolved inorganic carbon in water samples. Rapid Communications in Mass Spectrometry: An International Journal Devoted to the Rapid Dissemination of Up‐to‐the‐Minute Research in Mass Spectrometry 2009, 23(16), 2507-2510.

Point 11: L105 and 109: the authors used GF/F filters; their diameter is not as important as the pore size.

Response 11: Thanks for your point. We erased diameter information of GF/F filters.

Point 12: L141: lyophilizing time is irrelevant; what matters is if the samples were dry.

Response 12: If the filter samples for pigment analysis are lyophilized for a long time, the pigment may be degraded. Therefore, we mentioned lyophilizing time for pigment sample.

Point 13: L150-151: should be v/v/v.

Response 13: Thanks for your points. We revised that sentence (Line: 141142).

Revision 13:

“The mobile phases were (A) methanol: acetonitrile: aqueous pyridine solution (50: 25: 25, v: v: v) and (B) methanol: acetonitrile: acetone (60: 20: 60, v: v: v).”

“The mobile phases were (A) methanol: acetonitrile: aqueous pyridine solution (50: 25: 25, v/v/v) and (B) methanol: acetonitrile: acetone (60: 20: 60, v/v/v).”

Point 14: L157: authors should state what other pigments they are referring to.

Response 14: We had mentioned in the results section that each marker pigment describing phytoplankton (Line: 149151). But we added this information in methodology section for accurate understanding.

Revision 14:

“Fucoxanthin is a marker pigment of diatoms, alloxanthin is a marker pigment of cryptophytes, lutein and Chl. b are marker pigments of chlorophytes, and zeaxanthin is a marker pigment of cyanobacteria [26].”

Jeffrey, S.W.; Vesk, M. Introduction to marine phytoplankton and their pigment signatures. In Phytoplankton Pigments in Oceanography: Guidelines to Modern Methods; Jeffrey, S.W.; Mantoura, R.F.C.; Wright, S.W. (ed.); UNESCO: Paris, 1997, pp. 37–84.

Results

Point 15: Table 1: erase periods from table titles. Specify what you mean by “EL.m”.

Response 15: It seems there was a problem in server. Very sorry for that. We checked very carefully when resubmitting in this time. And the information about “EL.m” was added below the Table 1.

Point 16: On this table and in the rest of the text, it is irrelevant to state exactly the day of maximum and minimum parameters, since it is -of course- the only day in the month it was sampled. This would be important if they sampled every day.

Response 16: We erased all sampling dates in Table. Thanks.

Point 17: The authors do not explain the lack of information about 2 weirs in Table 1.

Response 17: It was mentioned in 2.3 (Line: 122125) why information about two study sites (Samunjin bridge and Goryeng bridge) was lacking. But we revised that sentence more accurate. The two study sites could not be monitored physicochemical parameters such as inflow, discharge, and storage volume. Because there are located in upper stream far from representative weirs.

Revision 17:

“HRT was calculated at four study sites by dividing the total water storage capacity by the amount of inflow water, except for the upper stream sites (ND-2 and ND-3) far from Dalseong weir.”

“HRT was calculated at four study sites using the total water storage capacity and the amount of inflow water. The hydrological data from upper stream sites far from Dalseong weir (ND-2 and ND-3), which are not representative sites in front of weir, could not be shown.”

Point 18: All the text: space is missing between the number and °C.

Response 18: As a result of looking up international Système international d'unités (SI), there is no space between numbers and symbols that are not alphabet. In the case of temperature, space may be used between numbers and units. But, in most scientific papers, space is not allowed. Accordingly, we did not put space between the number and °C in this manuscript. Thanks.

Point 19: P7L16: the authors mention a “relatively low value”; they need to state these values were relatively low compared to what.

Response 19: We revised this sentence. Thanks for the Reviewer #4’s point (Line: 219220).

Revision 19:

“They decreased after April 27th, and then remained relatively low value until June 25th and then increased sharply after July.”

“They decreased after April 27th, and then remained relatively low value until June 25th compared to the samples on April 27th and then increased sharply after July.”

Point 20: Table 2: same as Table 1: the day of the month is irrelevant, and this information is not useful but makes the table difficult to read.

Response 20: We erased Table 2, instead we represented the concentration of nutrients in Figure 2.

Point 21: P9L7-12: These data should be in a table for better visualization. And again, the day only reflects the day of sampling.

Response 21: The figure of seasonal and spatial variations in primary production was in manuscript (Figure 3). And more detailed figure of primary production in all sampling time was demonstrated in revised Figure S3. Thanks.

Point 22: P9L16: Clarify what you refer to with “They were…”.

Response 22: We revised this sentence. Thanks for the Reviewer #4’s point (Line: 236).

Revision 22:

“They were significantly lower in autumn than in spring (p < 0.01) and summer (p < 0.01).”

“It was significantly high in spring (p < 0.01) and summer (p < 0.01) compared to autumn.”

Point 23: P10L22: this sentence should be re-written.

Response 23: We revised this sentence. Thanks for the Reviewer #4’s point (Line: 242).

Revision 23:

“Phytoplankton have marker pigments according to different their classes.”

“Phytoplankton have different marker pigments according to their classes.”

Point 24: P10L37-60: there are several grammar mistakes. The whole paragraph should be improved with special attention to the word “contribution” (using singular or plural as needed and trying to avoid so much repetition).

Response 24: We corrected the word “contribution”, in addition, the repeated word was erased. Thanks (Line: 258, 262, 265, 269).

Revision 24:

“The relative contributions of phytoplankton community calculated by the CHEMTAX program showed a good correlation with cell numbers measured by microscopy, except for cryptophytes (Supplementary Figure S4).”

“The results calculated by the CHEMTAX program did not differ among the study sites (p > 0.05).”

“During the study period, the relative contribution of diatoms to the total phytoplankton classes ranged from 0 to 96% (46 ± 32%), cryptophytes from 2 to 64% (20 ± 16%), chlorophytes from 0 to 76% (17 ± 18%), and cyanobacteria from 0 to 88% (17 ± 24%) (Figure 5).”

“The relative contribution of diatoms was high from April 27th (89 ± 6.4%) to June 15th (52 ± 23%). It decreased to 5.5 ± 13% on August 8th, and they dominated again on September 6th, with a value of 85 ± 9.3%.”

Discussion

Point 25: P11L80: the authors refer to the proliferation of algal blooms, but they really mean cyanobacterial blooms.

Response 25: In the spring as well as in the summer, there were diatom blooms with primary production showing a value of over 1,000 mg C m-2 d-1. Therefore, this algal blooms do not mean only cyanobacterial blooms.

Point 26: P12L99: was A major, not an.

Response 26: Very thanks for checking carefully. We corrected that sentence (Line: 316).

Revision 26:

“Water temperature was an major factor that affects primary productivity in high TP conditions, but the effect of water temperature was found to decrease at low TP concentrations in Lake Geneva, Switzerland.”

“Water temperature was a major factor that affects primary productivity in high TP conditions, but the effect of water temperature was found to decrease at low TP concentrations in Lake Geneva, Switzerland.”

Point 27: P12L122-123: Nakdong River is unnecessarily repeated.

Response 27: We revised repeated word. Thanks (Line: 337342).

Revision 27:

“Approximately 60% of the TP load in the Nakdong River could be originated from non-point source pollution [25]. The water condition of the Nakdong River was the PO43- limited environment (Supplementary Figure S9) based on the criteria suggested by Justić et al. [41].”

“Approximately 60% of the TP load in the Nakdong River could be originated from non-point source pollutions [19]. The water condition of this study sites was the PO43- limited environment (Supplementary Figure S5) based on the criteria suggested by Justić et al. [42].

Point 28: P13L138: the relative contribution was.

Response 28: As the Reviewer #4’s point, we corrected that sentence. Thanks (Line: 356).

Revision 28:

“The relative contribution of cryptophytes were relatively higher in September and October compared to other seasons.”

“The relative contribution of cryptophytes was relatively higher in September and October compared to other seasons.”

Point 29: L142: Microcystis and Aphanizomenon were the dominant cyanobacterial genera. It is important to state differences, since “algae” or “microalgae” is only a functional and not a taxonomic group.

Response 29: This study focused on the identification of key factors controlling variability of primary production, providing only phytoplankton community composition calculated by the CHEMTAX program. Therefore, this study did not discuss phytoplankton species data based on microscopic observation. In addition, the detailed information of phytoplankton species at the same study sites and period was previously given in Kim et al. (2019).

Kim, S.; Chung, S.; Park, H.; Cho, Y.; Lee, H. Analysis of environmental factors associated with cyanobacterial dominance after river weir installation. Water (Switzerland) 2019, 11(6), 1163.

Point 30: L145: delete the word cyanobacteria.

Response 30: As the Reviewer #4’s point, we erased the word cyanobacteria (Line: 368). Thanks.

Revision 30:

“These genera are known to produce cyanobacteria toxins such as MCs.”

“These genera are known to produce toxins such as MCs.”

Point 31: L148: upper trophic levels, not positions.

Response 31: We corrected trophic positions to trophic levels (Line: 370).

Revision 31:

“MCs can be transported to upper trophic positions through aquatic food web”

“MCs can be transported to upper trophic levels through aquatic food web”

Point 32: P15l29: harmful cyanobacteria.

Response 32: We are sorry, but we could not understand what you mean.

Conclusions

Point 33: P16L30-42: The conclusions are only a repetition of results. This whole part should be improved: should this study be repeated annually to be aware of risks for the ecosystem and human population? Why is this important? What are the authors' recommendations? Why?

Response 33: Really thanks the Reviewer #4’s suggestion. We added some implication of our result and finding as follows (Line: 403404, 410413)

.

Revision 33:

“We investigated primary productivity and phytoplankton community composition in the Nakdong River where cHABs occur every year.”

“Therefore, to reduce HRT through water discharge could be one of solutions for controlling primary production including cHABs in a continuous weir system. This study provides useful information regarding to the key factors controlling primary production and occurrence of cHABs in a continuous weir system.”

References

Point 34: scientific names are not in italics.

Response 34: We corrected reference section. Thanks for checking carefully.

Suppl. Mat.

Point 35: Fig. S1: legend states cubic millimeters when it should be cubic meters.

Response 35: Very thanks. We corrected wrong caption of Figure S1.

Fig. S9

Point 36: probably should not be supplementary but in the main text.

Response 36: We transferred Figures showing the concentrations of nutrients (Figurs S3, S4, S5, and S6) to main text (revised Figure 2). Instead, we erased existing Table 2 representing the concentrations of nutrients, instead of Figure S9. Thanks.

Round 2

Reviewer 2 Report

In the first version, the authors did not include the figures and it was not possible to review the complete manuscript. The problem persists, and the authors do not check the manuscript prior submission. In the revised version, there two sections materials and methods, and the repetition of the same paragraphs in the distinct parts of the manuscript. The line numbering disappears after the page 8, and after the page 9 continues the page 1. It is not possible to review the manuscript if the authors do not pay the due attention to the edition.

From the scientific point of view the problem remains. The authors continue using sentences as the fucoxanthin is the pigment of the diatoms, or Chl b of the chlorophytes. This is not true as other algal groups have these pigments. For that reason, it is necessary to complement the study with microscopical observations.

The authors omit that information that was published in:

Kim, S.; Chung, S.; Park, H.; Cho, Y.; Lee, H. Analysis of environmental factors associated with cyanobacterial dominance after river weir installation. Water (Switzerland) 2019, 11(6), 1163.

The only comparison is the supplementary figure S4: In the text is reported “The relative contributions of phytoplankton community calculated by the CHEMTAX program showed a good correlation with cell numbers measured by microscopy, except for cryptophytes (Supplementary Figure S4). This finding indicates that the CHEMTAX program based on pigment analysis can accurately describe phytoplankton community composition in a freshwater environment.”

Please remember that you are not reporting the phytoplankton community composition. For example, you are only reporting the relative contribution of the fucoxanthin to the total chlorophyll a. The ratio of fucoxanthin when compared to the total chlorophyll a in a single cell varies according the physiology and environmental conditions. Then, more relative proportion of fucoxanthin does not mean more diatoms. The cells can increase the production of zeaxanthin in sunny days or summer, but it does not means to have more cells of the group containing zeaxanthin. CHEMTAX is only other complementary method to obtain information about the phytoplankton, but never an accurate method to describe the phytoplankton community composition in a freshwater environment.

line 85: Please remove: Mackey et al. a ratios.

Please do not repeat sentences:

line 53: The Nakdong River is the longest river in South Korea with a length of 525 km and watershed area 53 of 23,384 km2. It is one of the largest water resources for drinking, industry, and agriculture [15].

line 86: The Nakdong River is the longest river in South Korea with a length of 525 km and watershed area 53 of 23,384 km2. It is one of the largest water resources for drinking, industry, and agriculture [15].

method line 89:

Proliferation of cyanobacteria changes the color of the water, creates scums and odors, and produces harmful toxins (e.g. microcystins, MCs) that are hazardous to drink and can potentially affect aquatic organisms

This is again a repetition that is not a description of the methods.

line 69: Materials and methods

line 92: Materials and methods

Why do you repeat the section of the material and methods?

line 103: Water samples (approximately 4 L) from four water depths (100%, 50%, 12%, and 1% surface irradiances) The 1% depth of relative light intensity was determined as the euphotic depth. were collected using a Van Dorn sampler.

This has no sense.

line 144: .In-situ incubation experiment for primary productivity

Filters were preserved at –80℃ until analysis. Carbon uptake rates of phytoplankton were calculated using  the equation of Hama et al. 2.4. Phytoplankton pigment analysis using HPLC and application of the CHEMTAX program

This has no sense. Please check the manuscript before submission!!!

line 165:  Fucoxanthin is a marker pigment of diatoms, alloxanthin is a marker pigment of cryptophytes, lutein and Chl. b are marker pigments of chlorophytes, and zeaxanthin is a marker pigment of cyanobacteria [26].

The authors continue ignoring my previous comments. This is fake. Fucoxanthin is not an specific marker pigment of diatoms because other groups have fucoxanthin. For example, the dinoflagellate Karenia has fucoxanthin.

Chl. b is not the marker pigment of chlorophytes. For example, the dinoflagellate Lepidodinium has chlorophyll b.

line: 173 . However, quantitative and qualitative measurements of phytoplankton marker pigments have limitations in estimating the phytoplankton community and distribution.

Right. The pigments do not reveal the phytoplankton community because the pigments are not exclusive of an algal group and the production of an accesory pigment changes with the physiological status. Then, please show the data of the microscopical observations (other than Figure S4) and explain the method.

I cannot continue using the line numbering for the review because the authors have removed it after the page 8, and even worst, after the page 9 continues the page number 1.

Each reference is denoted with a number, but in the reference list there are 9 references between the references numbers 35 and 36.

It is not possible to continue the review of a manuscript where the authors do not pay attention to the edition with repeated paragraphs in distinct part of the text, with duplicate page numbers and lacking the line numbering.

In the next revised version, please check the edition prior submission and include the comparison between the microscopic observations and the results of the HPLC (other than the Figure S4). Please cite the dominat species in each group.

Author Response

Response to Reviewer 2 Comments

Reviewer #2

Point 1: In the first version, the authors did not include the figures and it was not possible to review the complete manuscript. The problem persists, and the authors do not check the manuscript prior submission. In the revised version, there two sections materials and methods, and the repetition of the same paragraphs in the distinct parts of the manuscript. The line numbering disappears after the page 8, and after the page 9 continues the page 1. It is not possible to review the manuscript if the authors do not pay the due attention to the edition.

Response 1: We sincerely apologize that the same problems (not include figure, omitted line number) occurred again in the journal submission system. These problems seemed to occur in the process of formatting change when we submitted the first version. Unfortunately, it seemed to occur again when we submitted revised manuscript. Definitely we submitted the all figures and line number on the manuscript twice. However, we think the submission system has some problems to convert the original manuscript to the formatted version. If you can check the original manuscript we submitted, you may recognize all figures and line numbers in our submitted versions. In addition, we noticed this problem to assistant editor by e-mail. This time, we directly revised on the formatting changed file which we received from the journal system. We really hope the current manuscript is going to be transferred without any problem. Anyhow we are very sorry for the inconvenience you had in the reviewing manuscript.

Point 2: The authors omit that information that was published in:

Kim, S.; Chung, S.; Park, H.; Cho, Y.; Lee, H. Analysis of environmental factors associated with cyanobacterial dominance after river weir installation. Water (Switzerland) 2019, 11(6), 1163.

Response 2: We added this reference in our revised manuscript.

Point 3: From the scientific point of view the problem remains. The authors continue using sentences as the fucoxanthin is the pigment of the diatoms, or Chl b of the chlorophytes. This is not true as other algal groups have these pigments. For that reason, it is necessary to complement the study with microscopical observations.

The only comparison is the supplementary figure S4: In the text is reported “The relative contributions of phytoplankton community calculated by the CHEMTAX program showed a good correlation with cell numbers measured by microscopy, except for cryptophytes (Supplementary Figure S4). This finding indicates that the CHEMTAX program based on pigment analysis can accurately describe phytoplankton community composition in a freshwater environment.”

Please remember that you are not reporting the phytoplankton community composition. For example, you are only reporting the relative contribution of the fucoxanthin to the total chlorophyll a. The ratio of fucoxanthin when compared to the total chlorophyll a in a single cell varies according the physiology and environmental conditions. Then, more relative proportion of fucoxanthin does not mean more diatoms. The cells can increase the production of zeaxanthin in sunny days or summer, but it does not means to have more cells of the group containing zeaxanthin. CHEMTAX is only other complementary method to obtain information about the phytoplankton, but never an accurate method to describe the phytoplankton community composition in a freshwater environment.

Response 3: We absolutely agree with your opinion. Microscopic observation provides accurate phytoplankton species composition which is very important information. Microscopic analysis has a great advantage to identify phytoplankton species and the exact cell abundance of each species. However, this study tries to roughly classify phytoplankton class abundances using pigment analysis in comparison to microscopic observation data which was previously reported (Kim et al., 2019). The CHEMTAX program based on pigment composition was just tested to compare the microscopic data set and phytoplankton class composition. So we revised “phytoplankton community composition” to “phytoplankton class abundance” in the revised manuscript as your helpful comment.

Revision 3:

Line 18: we investigated spatio-temporal variations in P.P. and phytoplankton classes in a continuous weir system in the Nakdong River once or twice a month from April to October 2018.

Line 20: Relative abundance of the phytoplankton classes were calculated by the CHEMTAX program based on pigment analysis using a high-performance liquid chromatography (HPLC).

Line 25: The relative abundance of cyanobacteria (mainly Microcystis sp.) was positively correlated with water temperature (r = 0.79, p < 0.01) and hydraulic retention time (HRT, r = 0.67, p < 0.01),

Line 158-159: Mackey et al. [31] developed the CHEMTAX program to assess the abundance of each phytoplankton class using a steepest descent algorithm and factor analysis based on pigment/Chl. a ratios.

Line 162: In this study, the relative abundance of phytoplankton classes was calculated by the CHEMTAX program.

Line 270: 3.4. Relative abundance of phytoplankton classes calculated by the CHEMTAX program

Line 271: The relative abundance of phytoplankton classes

Line 273-274: This finding indicated that the CHEMTAX program based on pigment analysis can accurately describe abundance of phytoplankton classes in a freshwater environment.

Line 276: Significant seasonal variations in relative abundance of phytoplankton classes

Line 279-280: The relative abundance of diatoms

Line 282: The relative abundance of diatoms

Line 286: The relative abundance of cryptophytes

Line 288: The relative abundances of chlorophytes and cyanobacteria

Line 290: The abundance of chlorophytes

Line 291: Seasonal variations in cyanobacteria abundance

Line 378: The relative abundance of diatoms showed a negative relationship

Line 405: The relative abundance of cyanobacteria

Line 415: On the contrary, the relative abundance of cyanobacteria

Point 4: line 85: Please remove: Mackey et al. a ratios.

Please do not repeat sentences:

line 53: The Nakdong River is the longest river in South Korea with a length of 525 km and watershed area 53 of 23,384 km2. It is one of the largest water resources for drinking, industry, and agriculture [15].

line 86: The Nakdong River is the longest river in South Korea with a length of 525 km and watershed area 53 of 23,384 km2. It is one of the largest water resources for drinking, industry, and agriculture [15].

method line 89:Proliferation of cyanobacteria changes the color of the water, creates scums and odors, and produces harmful toxins (e.g. microcystins, MCs) that are hazardous to drink and can potentially affect aquatic organisms

This is again a repetition that is not a description of the methods.

line 69: Materials and methods

line 92: Materials and methods

Why do you repeat the section of the material and methods?

Response 4: All the above problems were caused by the conversion process in the journal system. We couldn’t understand why these problems occurred. We resubmitted the revised manuscript after confirming the revised manuscript. We really hope to transfer our revised manuscript without any problem.

Point 5: line 103: Water samples (approximately 4 L) from four water depths (100%, 50%, 12%, and 1% surface irradiances) The 1% depth of relative light intensity was determined as the euphotic depth. were collected using a Van Dorn sampler.

This has no sense.

Response 5: We deleted water volume information (approximately 4 L) in the sentence.

Point 6: line 144: In-situ incubation experiment for primary productivity

Filters were preserved at –80℃ until analysis. Carbon uptake rates of phytoplankton were calculated using the equation of Hama et al. 2.4. Phytoplankton pigment analysis using HPLC and application of the CHEMTAX program

This has no sense. Please check the manuscript before submission!!!

Response 6: We could not find the above sentence on line 144. However, it seems to be a part of conversion problem in the system.

Point 7: line 165: Fucoxanthin is a marker pigment of diatoms, alloxanthin is a marker pigment of cryptophytes, lutein and Chl. b are marker pigments of chlorophytes, and zeaxanthin is a marker pigment of cyanobacteria [26].

The authors continue ignoring my previous comments. This is fake. Fucoxanthin is not an specific marker pigment of diatoms because other groups have fucoxanthin. For example, the dinoflagellate Karenia has fucoxanthin.

Chl. b is not the marker pigment of chlorophytes. For example, the dinoflagellate Lepidodinium has chlorophyll b.

Response 7: This sentence means that the marker pigments are relatively abundant pigments for each phytoplankton class, even though some marker pigments are commonly contained by a few phytoplankton species. As your suggestion, this sentence may lead to the misunderstanding that diatoms have only fucoxanthin, and others as well. We deleted this entire sentence.

Point 8: line: 173 . However, quantitative and qualitative measurements of phytoplankton marker pigments have limitations in estimating the phytoplankton community and distribution.

Right. The pigments do not reveal the phytoplankton community because the pigments are not exclusive of an algal group and the production of an accesory pigment changes with the physiological status. Then, please show the data of the microscopical observations (other than Figure S4) and explain the method.

Response 8: The microscopic data was already published (Kim et al., 2019). So we just cited those data in this study.

Point 9: I cannot continue using the line numbering for the review because the authors have removed it after the page 8, and even worst, after the page 9 continues the page number 1.

Response 9: The same problem occurred in submission system.

Point 10: Each reference is denoted with a number, but in the reference list there are 9 references between the references numbers 35 and 36.

Response 10: This is the print screen of the manuscript we downloaded. We could not find 9 references between the references numbers 35 and 36. We are not sure why these problems occurred

Point 11: It is not possible to continue the review of a manuscript where the authors do not pay attention to the edition with repeated paragraphs in distinct part of the text, with duplicate page numbers and lacking the line numbering.

Response 11: These problems are not our mistakes. So we really hope to transfer our submitted manuscript to reviewers without any errors in the system.

Point 12: In the next revised version, please check the edition prior submission and include the comparison between the microscopic observations and the results of the HPLC (other than the Figure S4). Please cite the dominant species in each group.

Response 12: We sincerely apologize for this kind of errors. The data on species through microscopic analysis have already been published (Kim et al., 2019), and we cited the previous data and report for the identification of dominant species, comparing the microscopic observations and the pigment analysis data sets at the class level. We added the sentence explaining dominant species in phytoplankton class by quoting the reference. Thanks you so much for your very constructive and kind comments.

Revision 12: (Line 277-279) During the study period, Aulacoseira sp., Cryptomonas sp., Eudorina sp., and Microcystis sp. were dominant species in diatoms, cryptophytes, chlorophytes, and cyanobacteria, respectively [35, 36].

“35. Korea Water Resources Corporation. A study on causes and prediction of cyanobacterial blooms in Korea’s 3 major rivers (Report); Korea. 2018, 68–84.”

“36. Kim, S.; Chung, S.; Park, H.; Cho, Y.; Lee, H. Analysis of environmental factors associated with cyanobacterial dominance after river weir installation. Water (Switzerland) 2019, 11(6), 116

Reviewer 3 Report

I thank the authors for having attended the recommendations made. I have some other observations

Author Response

Response to Reviewer 3 Comments

Reviewer #3

I thank the authors for having attended the recommendations made. I have some other observations

Response: Thanks for your comments. We revised the manuscript as your suggestion. Thanks.

Introduction

Point 1: It’s necessary to clearly define the elements under study in the introduction:

- What is Primary Productivity, a bibliographic reference at least.

- What is a cHABs, a bibliographic reference at least.

Response 1: Thanks for your comments. We added the reference explaining primary productivity and cHABs in introduction.

Revision 1:

Line 35~36: Primary productivity represents phytoplankton biomass times their growth rate, which is defined as the mass of fixed carbon [1].

Line 49~51: The cHABs are increasing recently, and these occur mainly in eutrophic rivers, lakes, and reservoirs in summer when water temperature is over 25℃ [14].

  1. Cloern, J.E.; Foster, S.Q.; Kleckner, A.E. Phytoplankton primary production in the world’s estuarine-coastal ecosystems. Biogeosciences 2014, 11(9), 2477–2501.
  2. Paerl, H.W.; Huisman, J. Climate change: A catalyst for global expansion of harmful cyanobacterial blooms. Environ. Microbiol. Rep. 2009, 1(1), 27–37.

Point 2: Line 53-59. I insist that these lines should go in Methodology, in study sites.

Response 2: We agreed to your opinion and transferred the sentence about the overview description of the Nakdong River to the methodology part. The Nakdong River has the consecutive weirs in the main stream. This artificial environment is uncommon in natural environment. Accordingly, the description of the weir construction in the Nakdong River was left in the introduction. Thanks.

Revision 2:

Line 56~64: The Nakdong River is the longest river in South Korea with a length of 525 km and watershed area of 23,384 km2. The Korean government conducted the ‘Four Major Rivers Project’ from 2009 to 2012 to solve repeated flooding and droughts [16]. Eight continuous weirs were constructed over approximately 200 km on the main stream. After construction of the weirs, there was an increase in HRT, water level, and storage volume [17]. In addition, total phosphorus (TP) concentrations decreased noticeably from 2001 to 2016, particularly after 2013 due to reinforcement of water quality regulations [18]. Since the 1990s, diatom blooms have been reported in spring and winter, and cyanobacterial blooms in summer [19]. After the weir construction, the cell density of diatoms decreased, while cyanobacteria abundance increased [20].

Line 73~75: The Nakdong River is one of the largest water resources for drinking, industry, and agriculture in South Korea [20]. The areas surrounding the middle and downstream are densely populated, and industrial complexes are concentrated near by the Nakdong River [21].

Materials and methods

Point 3: Line 85. The reference is not well written

Line 92. Study sites section is repeated

Response 3: We sincerely apologize that the same problems (not include figure, omitted line number) occurred again in the journal submission system. These problems seemed to occur in the process of formatting change when we submitted the first version. Unfortunately, it seemed to occur again when we submitted revised manuscript. Definitely we submitted the all figures and line number on the manuscript twice. However, we think the submission system has some problems to convert the original manuscript to the formatted version. If you can check the original manuscript we submitted, you may recognize all figures and line numbers in our submitted versions. In addition, we noticed this problem to assistant editor by e-mail. This time, we directly revised on the formatting changed file which we received from the journal system. We really hope the current manuscript is going to be transferred without any problem. Anyhow we are very sorry for the inconvenience you had in the reviewing manuscript.

Results

Point 4: Figure 2. I would like to see also the precipitation of the same days of observations

Response 4: The figures representing precipitation during the study period was shown in supplementary Figure S1.

The precipitation was high in early July and late August to early September. Unfortunately, sampling was not conducted in late July and early August. Therefore, the highest rainfall during the sampling period was in early September. Accordingly, average primary production at all study sites was the lowest on September 6th with the value of 42 ± 40 mg C m−2 d−1.

Point 5: Line 121-122. I don't see how the depth of light was obtained, I know it was obtained with a LiCOR device, but they don't explain the transformation of light values to Optical Depth values, that is to say, how do you determine the percentages of light depths?

Response 5: We first measured the light intensity at surface using LI-COR. And light intensity was measured to the bottom at 5 m depth intervals. For example, if the light intensity at surface is 1000 µmol m-2 sec-1 and the light intensity at 2 m depth is 500 µmol m-2 sec-1, 2 m is the water depth of 50% of surface light irradiance.

Point 6: On Page 10 line and page numbering are lost.

Response 6: The above problems were caused by the conversion process in the journal system. We couldn’t understand why these problems occurred. We resubmitted the revised manuscript after confirming the revised manuscript. We really hope to transfer our revised manuscript without any problem.

Point 7: At the beginning of page 10 is the importance of light availability, logically, on the surface the highest Primary Productivity due to availability of light.

Response 7: We tested the correlation between light intensity and primary production. But primary production was not significantly correlated with light intensity (p > 0.05). PI curves did not show significant correlation except for ND-1 (Pmax = 843.0) and ND-2 (Pmax = 892.0). In this study, it seems that various environment factors besides light intensity affected primary production besides. We determine that the PI curves were not discussed in manuscript. The figures of PI curve were shown as follows (We attatched PI curve figure. Please see the attatchment).

Discussion

Point 8: It’s said that the control in the hydraulic retention time can solve or reduce the presence of cHABs, so in this case, what happens to the temperature, will it stop being an active variable?

Response 8: Water temperature can affect the cHABs, and also the control of HRT causes water temperature change which is related to physiological activity of harmful algae.

Point 9: If temperatures are higher in Summer, why is Primary Productivity higher in Spring?

Response 9: This study explained that high water temperature affected the increase in contribution of cyanobacteria, not primary production. In spring, primary production was high, but the relative contribution of cyanobacteria was low.

Point 10: Secchi's disc is mentioned in results, but it is not mentioned either in the introduction or in the methodology in its part where it talks about the depth of light

Response 10: We agreed with your comments. We added the sentence about the secchi disk in the methodology part. Thanks a lot.

Revision 10:

Line 90~92: The secchi disk is a circular white disk with a diameter of 30 m. The depth of disappearance of disk was measured to evaluate the transparency of waters.

Reviewer 4 Report

All comments have been addressed. The paper now has been improved.

Author Response

Response to Reviewer 4 Comments

Reviewer #4

Point: All comments have been addressed. The paper now has been improved.

Response: We appreciate your positive and encouraging comments. We could improve our manuscript thanks to your suggestions and concerns. Very Thanks.

Round 3

Reviewer 2 Report

I cannot find the changes in the ms. Please activate the tracking changes in MS WORD. After the page 8 continues the page 0. Please solve these problems in edition of the revised version before sending the ms to the Reviewers.

Author Response

Response to Reviewer 2 Comments

Reviewer #2

Point 1: I cannot find the changes in the ms. Please activate the tracking changes in MS WORD. After the page 8 continues the page 0. Please solve these problems in edition of the revised version before sending the ms to the Reviewers.

Response 1: We activated the tracking change function and submitted. We corrected the page number.

Point 2: line 19: phytoplankton classes

line 21: Relative abundance of the phytoplankton classes

In the first version was reported "Relative contributions of the phytoplankton community". Now, classes have erroneously replaced this. What class? Diatom is not a class. The diatoms are divided into three classes: Coscinodiscophyceae, Fragilariophyceae, Mediophyceae

Your method is unable to distinct among the classes of diatoms (Coscinodiscophyceae, Fragilariophyceae, Mediophyceae). This is similar for other groups. Please do not use taxonomical ranks (family, order, class, phylum) because the pigment-based method do not allow to establish the taxonomical ranks.

Response 2: We removed the taxonomical ranks. And we revised them to the relative proportion of pigment based phytoplankton composition.

Point 3: line 26: The relative abundance of cyanobacteria

In the first version: The relative contribution of cyanobacteria

The authors ignore the comments of this Reviewer. The abundance is measured as cells per liter. A relative abundance is a proportion in the cell per liters, but your method cannot estimate the abundance of cyanobacteria. Your method estimate the relative proportion of a pigment typically found in cyanobacteria with respect to the chlorophyll a. The amounts of an accessory pigment with respect to the chlorophyll a is not a fixed value. It depends of the physiology of the cell or the environmental conditions. Please be correct and use "the relative proportion of the "pigment name" typically found in cyanobacteria with respect to chlorophyll a was..."

Response 3: According to your suggestion, we revised the term to the relative proportion of pigment based phytoplankton composition.

Point 4: line 34: Primary productivity represents phytoplankton biomass times their growth rate

The first sentence of the introduction has no sense.

Response 4: We deleted the first sentence in the revised manuscript.

Point 5: line 43: environmental conditions of lakes.

The introduction should focus on the phytoplankton in rivers. You are not studying lakes. Phytoplankton in rivers are subjected to high turbulence/turbidity and continuous changes. Phytoplankton in lakes are in more stable conditions.

Response 5: We tried to explain the effect of water level fluctuations in water environment. We revised this sentence and added the reference accounting for the effect water level fluctuations in river.

Revision 5: The significant change of the amplitude of water-level fluctuations, causing strong disturbances in aquatic environments [12, 13], such as phytoplankton community composition and densities by affecting the amount of suspended sediments, light availability, and diluting biomass [6, 11] (Line 40~43).

“13. Cha, Y.K.; Cho, K.H.; Lee, H.; Kang, T.; Kim, J.H. The relative importance of water temperature and residence time in predicting cyanobacteria abundance in regulated rivers. Water Res. 2017, 124, 11–19.”

Point 6: line 49: The cHABs are increasing recently, and these occur mainly in eutrophic rivers, lakes, and reservoirs in summer when water temperature is over 25℃

Please do not add new sentences. There are toxic blooms of cyanobacteria in Canada, and water temperature is never over 25ºC. It is erroneous to attribute the blooms to a temperature values as you did in the discussion.

Response 6: We deleted the sentence as your suggestion.

Point 7: I insist in my first review and the manuscript should begin the introduction in "The Nakdong River is the longest river in South Korea…" because you are unable to write a general introduction with sense.

Response 7: Other reviewers suggested this part moved to materials and methods. So we should accept that suggestion. Because it is belonging to the study site description.

Point 8: line 65: "Our objectives in this study were to understand the key environmental factors controlling spatio-temporal variability in primary production, and to identify environmental conditions causing cHABs in a continuous weir system in the Nakdong River."

Your objectives are: "to understand the key environmental factors controlling spatio-temporal variability in primary production, and to identify environmental conditions causing cHABs"

Response 8: We revised the sentence as your comment.

Revision 8: Our objectives in this study were to understand the key environmental factors controlling spatio-temporal variability in primary production, and to identify environmental conditions causing cHABs in the regulated water environment (Line 61~64).

Point 9: abstract: line 19: we investigated phytoplankton classes

line 21: Relative abundance of the phytoplankton classes was calculated

line 26: The relative abundance of cyanobacteria

Your objectives are not to investigate the phytoplankton composition, but large parts of the abstract are focused on this topic. Please report your objectives according to your results.

Response 9: We revised the term to the relative proportion of pigment based phytoplankton composition. We removed the phytoplankton composition in our research objective in abstract.

Revision 9:

Line 18~19: we investigated spatio-temporal variations in P.P. in a continuous weir system in the Nakdong River once or twice a month from April to October 2018.

Line 20~22: Relative proportion of pigment based phytoplankton composition was calculated by the CHEMTAX program based on pigment analysis using a high-performance liquid chromatography (HPLC).

Line 25: The relative proportion of pigment based cyanobacteria

Point 10: line 70-75

2.1. Study sites and sample preparation

The Nakdong River is one of the largest water resources for drinking, industry, and agriculture in South Korea [20]. The areas surrounding the middle and downstream are densely populated, and industrial complexes are concentrated near by the Nakdong River [21]. There are eight weirs in the main stream of the Nakdong river that were installed in 2012 as a part of the Four Major Rivers Project.

As reported in the previous reviews, this is a repetition; you have always introduce the river in the introduction. Just described your sampling sites.

Please begin in line 75.

Response 10: We divided the subtitles to 2.1. Study sites and 2.2. Sample collection, separately.

Point 11: Figure 1: Anomalous scale bar: 3.5 7 14 21. Why multiples of 3.5 Km?

Why the legend of the map reported "Water samples were collected from April to 83 October 2018 once or twice a week."

A map shows a location, not the sampling period to be explained in M&M.

Response 11: The scale bar was corrected. Thanks for your comment. We also corrected the caption of map.

Point 12: line 89: In the revised version has been added: “The secchi disk is a circular white disk with a diameter of 30 m.”

Are you sure that you manage with a enormous disk of 30 m in diameter?

The Secchi disk used in marine waters of 30 cm in diameter and white. In freshwaters, it is used a disk of 20 cm with black and white sectors. You have used the wrong Secchi disk.

Response 12: Thank you for your so helpful comment.

Point 13: line 70. 2.1. Study sites and sample preparation

line 85: Light intensity was measured at 0.5 m interval water

The measurements of the light is not a study site, and it is not sample preparation. Please use a correct title heading for the section.

line 122: 2.3. Meteorological and physicochemical characteristics

Light is not related to meteorology or a physical characteristic?

Response 13: The subtitle was changed to sample collection because we collected water samples using light intensity data with the water depths.

Point 14: line 58: However, quantitative and qualitative measurements of phytoplankton marker pigments have limitations in estimating the phytoplankton class and distribution.

This is no for M&M. Please explain it in the introduction. Please explain what quantitative and qualitative are. What is the limitation in the distribution?

Response 14: In terms of other reviewer’s comment, the introduction of the CHEMTAX program was moved to materials and methods. And we revised that sentence.

Revision 14: However, quantitative analysis of phytoplankton marker pigments has limitation in estimating the dominant phytoplankton (Line 159-160).

Point 15: line 160: CHEMTAX program to assess the abundance of each phytoplankton class

Your data are no abundance. The abundance are the cells per liter, and you never give that kind data. I have no more patience. If you disagree, then report in the abstract the abundance of Microcystis as cells per liter. I have no more patience with the successive reviews of this manuscript. I would recommend rejecting this manuscript.

Response 15: We deleted the word of abundance in the entire manuscript. Instead of abundance, we used the relative proportion, as your comment.

Point 16: line 163: In this study, the relative abundance of phytoplankton classes was calculated by the CHEMTAX program using the surface pigment concentrations and previously published initial pigment/Chl. a ratios [32, 33].

The aim of your study should be reported iin the last paragraph of the introduction.

Please explain that you are previously published initial pigment/Chl. a ratios.

Response 16: We deleted this sentence and revised the following sentence.

Revision 16: The initial pigment/Chl. a ratios [33, 34] used in this study was described in Supplementary Table S1 (Line 164).

Point 17: line 276: This finding indicated that the CHEMTAX program based on pigment analysis can accurately describe abundance of phytoplankton classes in a freshwater environment

This should be in discussion, not results. If the CHEMTAX program based on pigment analysis can accurately describe abundance, please provide the abundance data as cells per Liter (that is the unit for abundance of phytoplankton).

Response 17: We deleted the word of abundance in the entire manuscript. Instead of abundance, we used the relative proportion, as your comment.

Point 18: line 279: Significant seasonal variations in relative abundance of phytoplankton classes were observed in diatoms (p < 0.01), cryptophytes (p < 0.05), chlorophytes (p < 0.01), and cyanobacteria (p < 0.01).

This is result, then please explain the seasonal variations in the abudance.

Response 18: We had explained seasonal variations in relative proportions of pigment based phytoplankton composition, and deleted the word of abundance in the entire manuscript.

Point 19: line 280: During the study period, Aulacoseira sp., Cryptomonas sp., Eudorina sp., and Microcystis sp. were dominant species in diatoms, cryptophytes, chlorophytes, and cyanobacteria, respectively.

What species? If you are unable to identify the species, then please provide micrographs as supplementary data.

Response 19: We just cited the reference.

Point 20: line 285-295 and the rest of the text. Please do not use the term abundance for the relative proportion of an accessory pigment with respect to Chl a.

Response 20: We deleted the word of abundance in the entire manuscript. Instead of abundance, we used the relative proportion of pigment based phytoplankton composition, as your comment.

Point 21: line 388: In this study, Microcystis sp. was the dominant genera when cHABs occurred in the Nakdong River. The Korean alert system of algal blooms, as criteria for cyanobacteria, have three levels which are Caution (500 cells mL−1), Warning (5,000 cells mL−1), and Outbreak (1,000,000 cells mL−1) [48]. The exceeding 5,000 cells mL−1 of cyanobacteria occurred 69% of all the study sites in summer (from June 391 to August), and the highest cyanobacterial cell density was 1,264,052 cells mL−1 at Hapcheon-Changnyeong weir in August 2018 [49]. In addition, the average cyanobacteria cell density was 694,667 cells mL−1 at the Dalseong weir, followed by 453,283 cells mL−1 at the Hapcheon-Changnyeong weir in summer [36]. These genera are known to produce toxins such as MCs.

First, how many genera of cyanobacteria? You only cited one genus.

"Microcystis sp. was the dominant genera" Microcystis sp. is a species, not a genus. The genus is Microcystis, not Microcystis sp. Please remember that genus is singular and genera is plural.

This sentence should be in the introduction and you have to explain your new contribution because the main data have been already published.

Response 21: Good comment. We changed genera to genus. This is important data previously published. And we used this data to discuss our study.

Revision 21: This genus is known to produce toxins such as MCs (Line 417).

Point 22: line 432 5. Conclusions

line 433: We investigated primary productivity and phytoplankton community composition in the Nakdong River

The phytoplankton community composition is not reported. There is not a list of the species of phytoplankton, and there is no data on the abundance of the species.

Response 22: We revised the phytoplankton community composition to the relative proportion of pigment based phytoplankton composition.

Point 23: line 436 Harmful algae (Microcystis sp.) biomass increased

The method to estimate the biomass of Microcystis is not reported in M&M. There is no any information on the biomass in the result section. If you have no measure the biomass, your conclusion cannot be the biomass.

Response 23: We deleted the word of biomass and revised this sentence.

Revision 23: The relative proportion of pigment based harmful algae (Microcystis sp.) increased due to the enhanced HRT and elevated water temperature, resulting in the occurrence of cHABs in a continuous weir system in summer (Line 458-459).

Point 24: line 438: A short HRT as well as limited light availability caused by high turbidity inhibited phytoplankton primary production despite the high concentrations of PO43- as a result of heavy rainfall inflow in September 6th.

What is a short HRT? Please provide a number of days that is considered short. What is the limit between short and large?

Response 24: The HRT on September 6th was less than 1 day which is relatively shorter than other sampling dates.

Point 25: line 440: Therefore, to reduce HRT through water discharge could be one of solutions for controlling primary production including cHABs in a continuous weir system

What value of HRT (measured in days) to control cHABs in your study?

Response 25: We think it depends on the environmental conditions, especially precipitation.

Reviewer 3 Report

I thank the authors for the work done a second time on their manuscript. I think they have improved the manuscript substantially and I acknowledge the effort.

I still do not understand why there is an extra document with figures and tables; they are furthermore important figures and tables to understand the text, which is not comfortable for the reader to have to go from one manuscript to another. Considering the work done by the authors, I propose a minor revision, as long as they can synthesize all those extra graphics in the same manuscript. The purpose of a manuscript is to synthesize the research carried out and the challenge is to clearly convey ideas and results.

I make a couple of observations below.

  • Lines 56-62 of the introduction and lines 73-76 of the methodology are repeated, It is the third time that I recommend it
  • The introduction has improved in the first two paragraphs, could talk in a third paragraph about examples of research on dams and lakes, in specific research on Primary Production.

Author Response

Response to Reviewer 3 Comments

Reviewer #3

Point 1: I thank the authors for the work done a second time on their manuscript. I think they have improved the manuscript substantially and I acknowledge the effort.

Response 1: Our manuscript could be improved thanks to your comments. We revised the manuscript as your suggestions. We sincerely hope our current manuscript is going to be accepted.

Point 2: I still do not understand why there is an extra document with figures and tables; they are furthermore important figures and tables to understand the text, which is not comfortable for the reader to have to go from one manuscript to another. Considering the work done by the authors, I propose a minor revision, as long as they can synthesize all those extra graphics in the same manuscript.

Response 2: In terms of your suggestion, we moved all supplementary figures to the main text. However, we would like to remain the tables because they are basic and reference data.

Point 3: The purpose of a manuscript is to synthesize the research carried out and the challenge is to clearly convey ideas and results.

Response 3: We revised the objective of our study as follows.

Revision 3: Although many researches have been done to investigate primary productivity in freshwater environments, few previous studies have reported spatial and temporal variations in primary production, including cHABs, in a continuous weir. It is important to identify the effects of anthropogenic environmental changes on phytoplankton in a continuous weir system being used as water resources. Our objectives in this study were to understand the key environmental factors controlling spatio-temporal variability in primary production, and to identify environmental conditions causing cHABs in the regulated water environment (Line 57~63).

Point 4: Lines 56-62 of the introduction and lines 73-76 of the methodology are repeated, It is the third time that I recommend it

Response 4: Lines 56-62 of the introduction (The Nakdong River is the longest river in South Korea with a length of 525 km and watershed area of 23,384 km2. The Korean government conducted the ‘Four Major Rivers Project’ from 2009 to 2012 to solve repeated flooding and droughts [16]. Eight continuous weirs were constructed over approximately 200 km on the main stream. After construction of the weirs, there was an increase in HRT, water level, and storage volume [17]. In addition, total phosphorus (TP) concentrations decreased noticeably from 2001 to 2016, particularly after 2013 due to reinforcement of water quality regulations [18].), and lines 73-76 of the methodology (The Nakdong River is one of the largest water resources for drinking, industry, and agriculture in South Korea [20]. The areas surrounding the middle and downstream are densely populated, and industrial complexes are concentrated near by the Nakdong River [21].) were different sentences.

But we transferred the paragraph about the Nakdong River from introduction to methodology section, as your suggestion. Thanks (Line 66~76).

Point 5: The introduction has improved in the first two paragraphs, could talk in a third paragraph about examples of research on dams and lakes, in specific research on Primary Production.

Response 5: We added the previous researches about primary productivity in freshwater environments in third paragraph. Thanks.

Revision 5: Anthropogenic factor such as excessive nutrients input affected the increase in primary productivity [8, 9]. In addition, climate change and long HRT consistently accelerate the overgrowth of cyanobacteria [13]. Lee et al. [5] suggested that the key factors controlling primary productivity were HRT and light conditions in Lake Chungpyeong, and phytoplankton physiological activity influenced by zooplankton grazing rate in Lake Paldang, Korea. Jia et al. [9] found that primary productivity was significantly correlated with dissolved total nitrogen, silicon/phosphorus, and dissolved inorganic carbon/phosphorus in the Gan River, and ammonium in Lake Poyang, China. (Line 51~57).
